# Probabilistic inference of the genetic architecture underlying functional enrichment of complex traits

Marion Patxot [1,11], Daniel Trejo Banos [1,11], Athanasios Kousathanas [1,11], Etienne J. Orliac[2], Sven E. Ojavee [1], Gerhard Moser [3], Alexander Holloway[1], Julia Sidorenko [4], Zoltan Kutalik [1,5,6], Reedik Mägi[7], Peter M. Visscher [4], Lars Rönnegård [8,9] & Matthew R. Robinson [10 ✉]

We develop a Bayesian model (BayesRR-RC) that provides robust SNP-heritability estimation, an alternative to marker discovery, and accurate genomic prediction, taking 22 seconds per iteration to estimate 8.4 million SNP-effects and 78 SNP-heritability parameters in the UK Biobank. We find that only ≤10% of the genetic variation captured for height, body mass index, cardiovascular disease, and type 2 diabetes is attributable to proximal regulatory regions within 10kb upstream of genes, while 12-25% is attributed to coding regions, 32–44% to introns, and 22-28% to distal 10-500kb upstream regions. Up to 24% of all cis and coding regions of each chromosome are associated with each trait, with over 3,100 independent exonic and intronic regions and over 5,400 independent regulatory regions having ≥95% probability of contributing ≥0.001% to the genetic variance of these four traits. Our open-source software (GMRM) provides a scalable alternative to current approaches for biobank data.

[1] Department of Computational Biology, University of Lausanne, Lausanne, Switzerland. [2] Scientific Computing and Research Support Unit, University of Lausanne, Lausanne, Switzerland. [3] Australian Agricultural Company Limited, Brisbane, QLD, Australia. [4] Institute for Molecular Bioscience, University of Queensland, Brisbane, QLD, Australia. [5] University Center for Primary Care and Public Health, Lausanne, Switzerland. [6] Swiss Institute of Bioinformatics, Lausanne, Switzerland. [7] Estonian Genome Center, Institute of Genomics, University of Tartu, Tartu, Estonia. [8] School of Technology and Business Studies, Dalarna University, Falun, Sweden. [9] Department of Animal Breeding and Genetics, Swedish University of Agricultural Sciences, Uppsala, Sweden. [10] Institute of Science and Technology Austria, Klosterneuburg, Austria. [11] These authors contributed equally: Marion Patxot, Daniel Trejo Banos, Athanasios Kousathanas. ✉email: matthew.robinson@ist.ac.at

As whole-genomes are collected for hundreds of thousands of individuals, we require regression methods that are not only computationally efficient, but which also provide improved inference. Rather than relying on subsets of the SNPs, methods should fully utilise the data, exploiting computational power to facilitate discovery of additional genomic regions, to improve understanding of the genomic architecture of common disease, and to provide more informative genomic prediction.

For example, when estimating the proportion of phenotypic variance attributable to different categories of genetic markers (the SNP-heritability, $h^2_{SNP}$ of a genomic region), recent studies[1–4] highlight the importance of accounting for minor allele frequency (MAF) and LD structure of the genomic data. Generally, assessment of the relative contribution of different genomic regions is currently made assuming that markers within a category all contribute to the variance, with enrichment defined as the estimated share of the variance explained divided by its expected share[5,6]. However ideally, the estimated distribution of marker effects for each category would be directly obtained, accounting for MAF and LD structure and allowing for some of the marker effects to be zero, as this would yield a better understanding of the polygenicity of genomic effects across different genomic annotation groups.

Furthermore, statistical inference usually follows a multi-step approach. Current mixed-linear association models such as those implemented in the software fastGWA[7], BoltLMM[8] and REGENIE[9], use a two-step approach, first estimating the variance contributed by the SNP markers without the use of MAF-LD-annotation information, and then estimating the marker effect sizes one-by-one as fixed effects in a second step[7,8,10]. Following this initial mixed-model association step, statistical inference (variance components, fine mapping and risk prediction) is then typically conducted on the summary statistics generated. The advantage of a multi-step approach is that large sample size can be easily obtained through meta-analyses, combining summary statistics from different studies and avoiding the need for individual-level data sharing. However, as large-scale biobank data is increasingly available, methods that provide joint estimates of the marker effects in a single step by estimating the effect sizes as random under flexible prior formulations may become beneficial as they: (i) can account for differences in the variance contributed across MAF, LD or annotation groups providing unbiased MAF-LD annotation-specific genetic effect size estimates and $h^2_{SNP}$ of different annotations, allowing for a contrasting of the genetic architectures of complex traits; (ii) give the probability that each marker, genomic region, annotation, gene-coding region, or SNP is associated with a phenotype, alongside the proportion of phenotypic variation contributed by each, yielding test statistics that describe the *gene* architecture of complex traits and the uncertainty over the estimates; and (iii) provide improved genomic prediction, whilst providing a posterior predictive distribution for each individual.

Here, we outline the fastest Bayesian penalised regression model to date, with a hybrid-parallel algorithm for analysing large-scale genomic biobank using a single command-line tool implemented in our grouped mixture regressions model (GMRM) software. We validate our approach in large-scale simulation study and provide an empirical example using four traits measured in both the UK Biobank and Estonian Biobank data.

## Results

**A Bayesian model for large-scale genomic data.** We derive a model that we call BayesRR-RC in Supplementary Note 1 and the "Methods" section, which is based on grouped effects with mixture priors, improving on the formulations of refs. [11–13]. Like these former methods, we consider a spike probability at zero (Dirac delta function), and a scale mixture of Gaussian distributions as a slab probability density. Unlike these models, we have genetic markers grouped into MAF-LD-annotation specific sets, with independent hyper-parameters for the phenotypic variance attributable to each group, so that the mixture proportions, the variance explained by the SNP markers, and the mixture constants are all unique and independent across SNP marker groups. This enables estimation of the phenotypic variance attributable to the group-specific effects, and differences in the underlying distribution of the $\boldsymbol{\beta}_\varphi$ effects among MAF-LD-annotation groups, with different degrees of sparsity. Assuming $N$ individuals and $p$ genetic markers, our model of an observed phenotype vector $\mathbf{y}$ is:

$$\mathbf{y} = \mathbf{1}\mu + \sum_{\varphi=1}^{\Phi} \mathbf{X}_\varphi \boldsymbol{\beta}_\varphi + \boldsymbol{\epsilon}, \qquad (1)$$

where there is a single intercept term $\mathbf{1}\mu$ and a single error term, a vector $(N \times 1)$ of residuals $\boldsymbol{\epsilon}$, with $\boldsymbol{\epsilon}|\sigma_\epsilon^2 \sim \mathcal{N}(0\mathbf{I}\sigma_\epsilon^2)$. An $N$ by $p$ matrix of single nucleotide polymorphism (SNP) genetic markers, centred and scaled to unit variance, which we denote as $\mathbf{X}_\varphi$. The effects are allocated into groups $(1, \ldots, \Phi)$. Each group has a set of model parameters $\Theta_\varphi = \{\boldsymbol{\beta}_\varphi, \pi_\varphi, \sigma_{G\varphi}^2\}$, with $\boldsymbol{\beta}_\varphi$ as a $p_\varphi \times 1$ vector of partial regression coefficients, where $\beta_{\varphi_j}$ is the effect of a 1 SD change in the $j$th covariate within the $\varphi$th group. The spike and slab prior, contains what is called a Dirac spike[14,15] for $\boldsymbol{\beta}_\varphi$, which induces sparsity in the model through a Dirac-delta at zero, excluding variables from the model by setting their coefficients to zero. A finite scale mixture of normal distributions centred at zero constitute the slab component. The slab shrinks the non-zero coefficients towards zero according to the slab's width, and by having a scale mixture of Gaussians, the distribution has heavier tails and can accommodate big and small effects[16]. Therefore, each $\beta_{\varphi_j}$ is distributed according to:

$$\beta_{\varphi_j} \sim \pi_{0\varphi}\delta_0 + \pi_{1\varphi}\mathcal{N}\left(0, \sigma_{1\varphi}^2\right) + \pi_{2\varphi}\mathcal{N}\left(0, \sigma_{2\varphi}^2\right) + \ldots + \pi_{L_\varphi\varphi}\mathcal{N}\left(0, \sigma_{L_\varphi\varphi}^2\right),$$
$$(2)$$

where for each SNP marker group $\{\pi_{0\varphi}, \pi_{1\varphi}, \ldots, \pi_{L_\varphi\varphi}\}$ are the mixture proportions and $\{\sigma_{1\varphi}^2, \sigma_{2\varphi}^2, \ldots, \sigma_{L_\varphi\varphi}^2\}$ are the mixture-specific variances proportional to

$$\begin{bmatrix} \sigma_{1\varphi}^2 \\ \vdots \\ \sigma_{L_\varphi\varphi}^2 \end{bmatrix} = \sigma_{G\varphi}^2 \begin{bmatrix} C_{1\varphi} \\ \vdots \\ C_{L_\varphi\varphi} \end{bmatrix},$$

with $\sigma_{G\varphi}^2$ the phenotypic variance associated with the SNPs in group $\varphi$, which, like all the other parameters, is estimated directly from the data. Here, we use 78 MAF-LD-annotation SNP marker groups. SNPs are partitioned into seven location annotations preferentially to coding (exonic) regions first, then to intronic regions, then to 1 kb upstream regions, then to 1–10 kb regions, then to 10–500 kb regions, then to 500–1 Mb regions. Remaining SNPs were grouped in a category labelled "others" and also included in the model so that variance is partitioned relative to these also. Thus, we assigned SNPs to their closest upstream region, for example if a SNP is 1 kb upstream of gene X, but also 10–500 kb upstream of gene Y and 5 kb downstream for gene Z, then it was assigned to be a 1 kb region SNP. This ensures that SNPs 10–500 kb and 500 kb–1 Mb upstream are distal from any known gene. We further partition upstream regions to experimentally validated promoters, transcription factor binding sites (tfbs) and enhancers (enh) using the HACER, snp2tfbs databases

(see "Code availability" section). All SNP markers assigned to 1 kb regions map to promoters; 1–10 kb SNPs, 10–500 kb SNPs, 500 kb–1 Mb SNPs are then split into enh, tfbs and others (unmapped SNPs) extending the model to 13 annotation groups (Supplementary Data 1). Within each of these annotations, we have three minor allele frequency groups (MAF ≤ 0.01, 0.01 < MAF ≤ 0.05, and MAF > 0.05), and then each MAF group is further split into two based on median LD score. This gives 78 non-overlapping groups for which our BayesRR-RC model jointly estimates the phenotypic variation attributable to, and the SNP marker effects within, each group. For each of the 78 groups, SNPs were modelled using five mixture groups with variance equal to the phenotypic variance attributable to the group multiplied by constants (0, 0.0001, 0.001, 0.01, 0.1).

One of the major limitations preventing the application of Bayesian approaches to large-scale genomic data is the view that the computation of a posterior distribution is too expensive. In Supplementary Note 2, we derive a Bulk Synchronous hybrid-parallel (BSP) Gibbs sampling scheme for large-scale genomic data that allows both the data and the compute tasks to be split within and across compute nodes in a series of message-passing interface (MPI) tasks. We extend previous sparse residual updating schemes by deriving sampling steps to utilise whole genome sequence or SNP genetic data stored in mixed binary/sparse-index representation (see Supplementary Note 2), reducing computational complexity of a single Gibbs step from $\mathcal{O}(n)$ to $\mathcal{O}(n_z)$, with $n_z$ the number of non-zero genotypes, as SNP-phenotype covariance estimation (dot product calculation) is conducted as a series of look-up tables. We provide publicly available open source software (GMRM) that requires as little as 22 s per MCMC sample to estimate 78 group-specific $h^2_{SNP}$ parameters, and the inclusion probabilities and effect sizes of 8,433,421 markers in 382,466 individuals on standard Intel Xeon CPU processors (see "Code availability" section, Supplementary Note 2).

**Simulation study.** We first compare the model performance of BayesRR-RC to existing approaches across 18 different genetic architectures. We randomly selected 40,000 unrelated UK Biobank individuals and used 596,741 imputed SNP markers from chromosomes 19 to 22. We randomly selected either 1000, 10,000 or 100,000 LD independent (LD $R^2 < 0.1$) causal SNP markers. For each SNP marker set, we then simulated effect sizes from a normal distribution with zero mean and variance of 0.1, 0.3 or 0.6 divided by the number of causal variants and $\propto N(0, [p(1-p)]^{-0.25})$, with $p$ the allele frequency (see "Methods" section). This simulates stronger effect sizes for rare variants in line with recent empirical estimates and we simulated ten replicate phenotypes for each of the nine different genetic architectures. We then additionally repeat each simulation, sampling the SNP marker effects this time from 13 different distributions, one for each of 13 different genomic annotation groups with different proportions of $h^2_{SNP}$ to create nine further different genetic architectures. We compare our BayesRR-RC model to the following statistical models: (i) a restricted maximum likelihood (REML) model implemented in the software BoltREML[17] with the same 78 MAF-LD-annotation groups enabling a direct comparison, (ii) a Haseman–Elston (HE) regression using the same 78 group model implemented in the software RHEmc[18], (iii) summary statistic linkage disequilibrium score regression (LDSC)[19], with LD scores calculated using the same data, and the same 78 non-overlapping annotations in a 78 component LDSC annotation model, and (iv) summary statistic SumHer[6] (LDAK) with the same 78 non-overlapping annotations.

We find that BayesRR-RC estimates the phenotypic variation attributable to different genomic annotation groups comparable

with the BoltREML model, with similar correlation of the estimated and simulated values within each simulation replicate (Fig. 1a). In comparison, RHEmc, which also uses individual-level data, yields estimates with lower correlation with the simulated value, but higher than both summary statistic approaches implemented in LDSC and Sumher (Fig. 1a). We calculate estimates of enrichment, defined as the proportion of $h^2_{SNP}$ attributable to the annotation divided by the proportion of SNPs mapping to the annotation (for bayesRR-RC, because there is sparsity in the SNP effects, we define enrichment as the proportion of SNPs in the model that map to the annotation, see "Methods" section) and we compare these to the true simulated value. Compared to other approaches, we find that BayesRR-RC gives a lower probability of false enrichment, calculated as the proportion of times within a simulation replicate that an annotation group was incorrectly assigned as having enrichment greater than 2 (Fig. 1b). Thus, BayesRR-RC provides accurate partitioning of genomic enrichment across the genome.

In Supplementary Note 3, we propose a posterior probability window variance (PPWV) approach[20], which provides a probabilistic determination of association of a given LD block, genomic window, gene, or upstream region, relative to the amount of phenotypic variation attributable to that window. Our PPWV approach determines the posterior inclusion probability that each region and each gene contributes at least 0.001% to the $h^2_{SNP}$, with theory outlined in Supplementary Note 3 suggesting well controlled FDR. We determine the ability of our PPWV approach to correctly localise an association to LD blocks (defined as groups of markers with LD $R^2 \geq 0.1$) that contain causal variants, and compare this to using LD to clump mixed-linear model association estimates obtained using the BoltLMM software (Fig. 2a). We find that a PPWV approach identifies associated LD blocks with higher probability as compared to clumped MLMA associations, for all genetic architectures, with the exception of simulated phenotypes with enrichment and low polygenicity, where the small numbers of relatively large effect size regions are better identified with a single-marker regression approach (Fig. 2a). Thus, BayesRR-RC provides an alternative to standard genome-wide association studies to localise SNP-phenotype associations at the regional level, especially for traits with high polygenicity.

We then also compare the prediction accuracy obtained in an independent sample when creating genomic predictors using (i) effect sizes estimated by BayesRR-RC, (ii) fixed-effect SNP effect sizes estimated in the MLMA approach implemented in bolt, and (iii) effect size estimates obtained from four different genomic prediction models proposed in a recent paper[21], implemented in the LDAK software, which are suggested to outperform all other current approaches. In comparison to the best LDAK predictor, we find that BayesRR-RC obtains similar or improved prediction accuracy across all genetic architectures, with greater prediction accuracy gains observed under genetic architectures where the SNP effect distributions differed across genomic annotations (Fig. 2b). We find that given sufficient power, BayesRR-RC can obtain or even exceed the theoretical expectation of prediction accuracy under ridge regression assumptions (Fig. 2b, see "Methods" section).

We then conduct a number of follow-up simulation studies. Recent work has highlighted differences in statistical model performance depending upon the relationship of SNP marker effect size, LD and MAF[1,3,4]. We explore the performance of our model in theory, with highly correlated genetic markers in Supplementary Note 4. We also conducted another large-scale, but well-powered, simulation study to explore the model performance of BayesRR-RC as compare to existing approaches

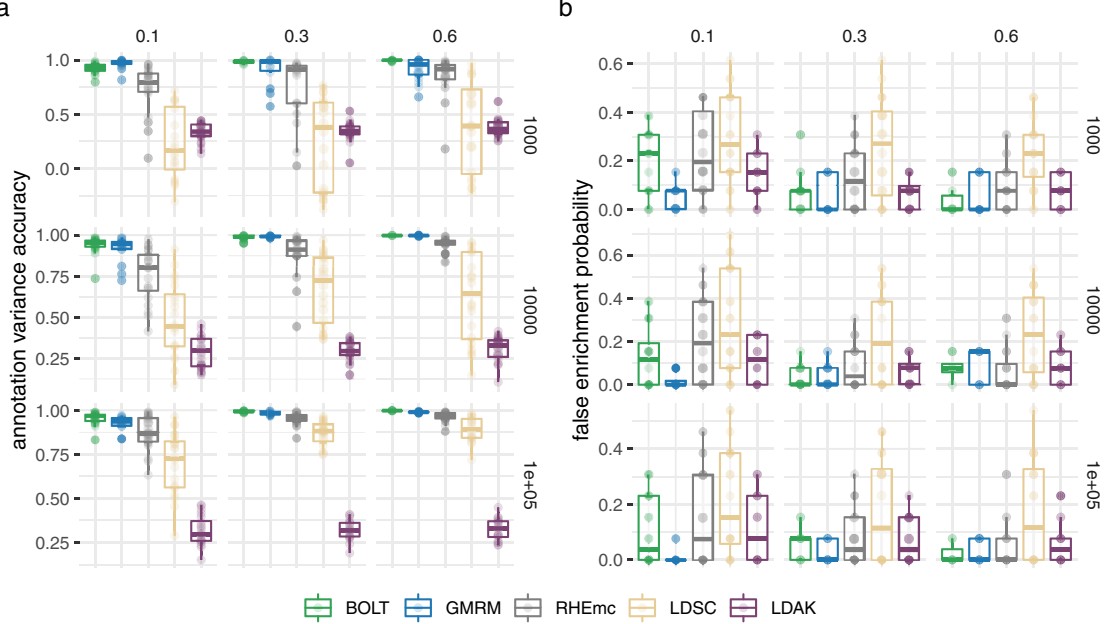

**Fig. 1 Simulation study for the performance of our BayesRR-RC model implemented in the GMRM software against existing approaches for variance component and genomic annotation enrichment estimation. a** Correlation of the simulated and estimated SNP heritability across 13 genomic annotation groups within each of 20 replicates for five different statistical models: a mixture of regression model with multiple group-specific variance components described in this work (GMRM), Haseman–Elston regression with annotation-specific relationship matrices implemented in the RHEmc software (RHEmc), a multiple group-specific variance component REML model implemented in the software bolt (BOLT), and two annotation summary statistic models implemented in the software LDSC and LDAK. The column facets give the simulated heritability and rows give the number of causal variants. **b** Probability of falsely assigning one of the 13 genomic annotation groups as explaining 2 times greater proportion of variance given the proportion of SNPs mapping to the annotation. The column facets give the simulated heritability and rows give the number of causal variants. Boxplots give the median with 25th and 75th percentile and 95% credible intervals for $n = 20$ simulation replicates in both panels.

across a wide range of 20 different effect size, LD, and MAF relationships as described in Supplementary Table 1. For the estimation of $h^2_{SNP}$ and the proportion of $h^2_{SNP}$ attributable to different annotation groups, we find that all statistical models other than BayesRR-RC are sensitive to the underlying generative genetic model, with no other approach providing consistent estimates across the 20 generative genetic models (Supplementary Fig. 1a). As in the previous simulation, BayesRR-RC estimates the variance attributable to different genomic regions on the correct scale, with higher correlation as compared with other approaches (Supplementary Fig. 1b), and this results in the estimated average effect size for each annotation group having high correlation with the simulated value (Supplementary Fig. 1c). Again, summary statistic approaches performed poorly for both variance component estimation (Supplementary Fig. 1b) and quantification of enrichment as compared to individual-level methods, often even incorrectly selecting the group of highest average effect size (Supplementary Fig. 1c).

We confirmed our genomic prediction results, finding that BayesRR-RC outperforms all methods implemented in the LDAK software across all generative models, with BayesRR-RC very marginally outperforming a single variance component BayesR model in the enrichment simulations of each of the 20 generative genetic models (Supplementary Fig. 2).

We further explored the ability of our PPWV approach to localise SNP-phenotype associations in the 20 generative models, by comparing the $z$-scores of the marker effect estimates from their true simulated value across the minor allele frequency spectrum (Supplementary Fig. 3) and the area under the precision-recall curve (AUPRC, Supplementary Fig. 4) for BayesRR-RC and a series of MLMA methods. We find that the

$z$-scores of the BayesRR-RC estimates are generally stable across generative genetic models and that the MLMA estimates have higher estimation error, especially when the causal variant is rare, or in high-LD with many other SNPs (Supplementary Fig. 3). We also find that our PPWV approach outperforms MLMA methods in their precision-recall curves across the range of genetic architectures (Supplementary Fig. 4). We confirmed that population stratification and relatedness are well-controlled for using a PPWV approach, as compared to an MLMA model with the leading PCs of the genomic data included (Supplementary Fig. 5). We compared the ability of our approach to identify candidate SNPs and to provide a probabilistic assessment of the most likely associated set of SNP markers. Finally, we show that our PPWV approach is analogous to the approach suggested in a recent paper (SuSiE[22]) of selecting credible sets of markers with high probability of association, finding that BayesRR-RC has higher power to localise associations to sets of SNP markers (Supplementary Fig. 6). The advantage of BayesRR-RC is also that assessment of associated regions is done genome-wide, with estimates obtained through simple summary of the posterior distribution instead of running numerous statistical models at different genomic regions. Taken together, these simulation results indicate that BayesRR-RC provides accurate estimates of the underlying effect size distribution for different genomic groups, yielding improved genomic prediction, across a wide range of different underlying generative genetic models.

**The genetic architecture of four complex traits in the UK Biobank.** We apply BayesRR-RC to cardiovascular disease outcomes (CAD), type-2 diabetes (T2D), body mass index (BMI) and height measured for 382,466 unrelated individuals from the

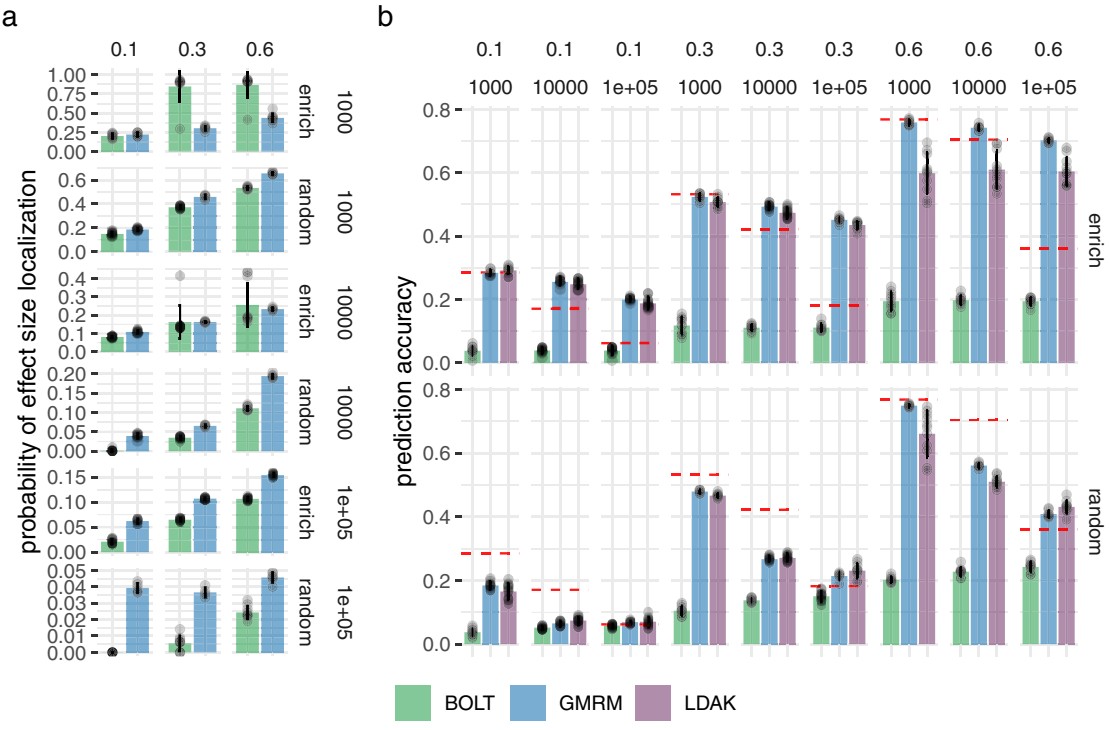

**Fig. 2 Simulation study for the performance of our BayesRR-RC model implemented in the GMRM software against existing approaches for localisation of associations and genomic prediction. a** Probability of detecting genomic regions containing simulated causal variants by a Bayesian regional fine-mapping approach (GMRM: blue) versus standard mixed linear model association (MLMA) testing (BOLT: green). The column facets give the simulated heritability and rows give the number of causal variants and whether the effect sizes differed across genomic annotation groups (enrich) or were randomly assigned (random). **b** Correlation of a genomic predictor and a phenotype in an independent sample when the genomic predictor is created from GMRM effects sizes (blue), MLMA effect sizes using BOLT (green), and the optimal effect sizes obtained from individual-level and summary statistic models implemented in the Mega-PRS LDAK approach (purple). The column facets give the simulated heritability and the number of causal variants. The row facets give whether the effect sizes differed across genomic annotation groups (enrich) or were randomly assigned (random). The red lines give the expected prediction accuracy based on ridge regression theory. Error bars show the SD in both panels.

UK Biobank data genotyped at 8,433,421 imputed SNP markers. These markers were selected as they overlap with the Estonian Genome Centre data (see "Methods" section) and have minor allele frequency >0.0002. We adjust each phenotype for age, sex, year of birth, genotype batch effects, UK Biobank assessment centre, and the leading 20 principal components of the SNP data. We conducted a series of convergence diagnostic analyses of the posterior distributions to ensure we obtained estimates from a converged set of four Gibbs chains, each run for 6000 iterations with a thin of five for each trait (Supplementary Figs. 7–10).

We find that 32–44% of the $h^2_{SNP}$ is attributable to intronic regions, 12–25% is attributable to exonic regions, 22–28% is attributable to markers 10–500 kb upstream of genes, with proximal (within 10 kb) promotors, enhancers and transcription factor binding sites cumulatively contributing <10% (Fig. 3b and Supplementary Fig. 11, with estimates summed across MAF and LD groups Table 1, and full results in Supplementary Data 2). The large contribution of exonic and intronic annotations to variation is in-line with the fact that these annotations account for ~40% of the total genome length. All four traits show the same pattern of group-specific variation, with the exception of height, where the proportion of $h^2_{SNP}$ attributable to exons is almost twice as large as the other phenotypes (Fig. 3b; Table 1 and Supplementary Fig. 11 and Supplementary Data 2). For all annotation groups in exons, introns, and within 500 kb of genes across all traits, ≥60% of the $h^2_{SNP}$ attributable to these groups is contributed by many thousands of common variants, each of small effect (Fig. 3b and Supplementary Figs. 11 and 12).

Our estimates compare similarly to those obtained by RHEmc and SumHer, but differ to those obtained by LDSC (Table 1 and Supplementary Data 3, 4, and 5 for full results). In addition to providing variance component estimates, our model facilitates assessment of differences in the underlying effect size distribution across annotation groups. For each group, we modelled the SNP effects as coming from a series of five Gaussian mixtures, and we find that at least 45% of the $h^2_{SNP}$ attributable to both introns and 500 kb upstream regions is underlain by many thousands of SNPs that on average each contribute 0.001% (estimates summed across MAF and LD groups in Fig. 3b and Supplementary Figs. 11 and 12). In contrast, the variance is spread more evenly across the mixtures for the other groups, implying that 10–500 kb upstream regions and introns are more polygenic than other groups. This is especially so for BMI where 35% of the $h^2_{SNP}$ is attributable to many thousands of intronic variants (Fig. 3 and Supplementary Fig. 12). Therefore, we find that the polygenicity of the genetic effects varies across different genomic regions, with remarkably consistent patterns across traits in the partitioning of $h^2_{SNP}$ across the genome.

Across traits, posterior mean effect sizes scale to their differences in $h^2_{SNP}$, and we find that exonic and intronic region effect sizes were higher than the rest of the genome, across all mixture groups, followed by 10–500 kb upstream regions (Fig. 3c). We find little evidence that SNPs located in proximal promotors, enhancers, and transcription factor binding sites within 10 kb of genes showed average effect sizes that were higher than SNPs located 1 MB away from genes, or those that were not mapped to

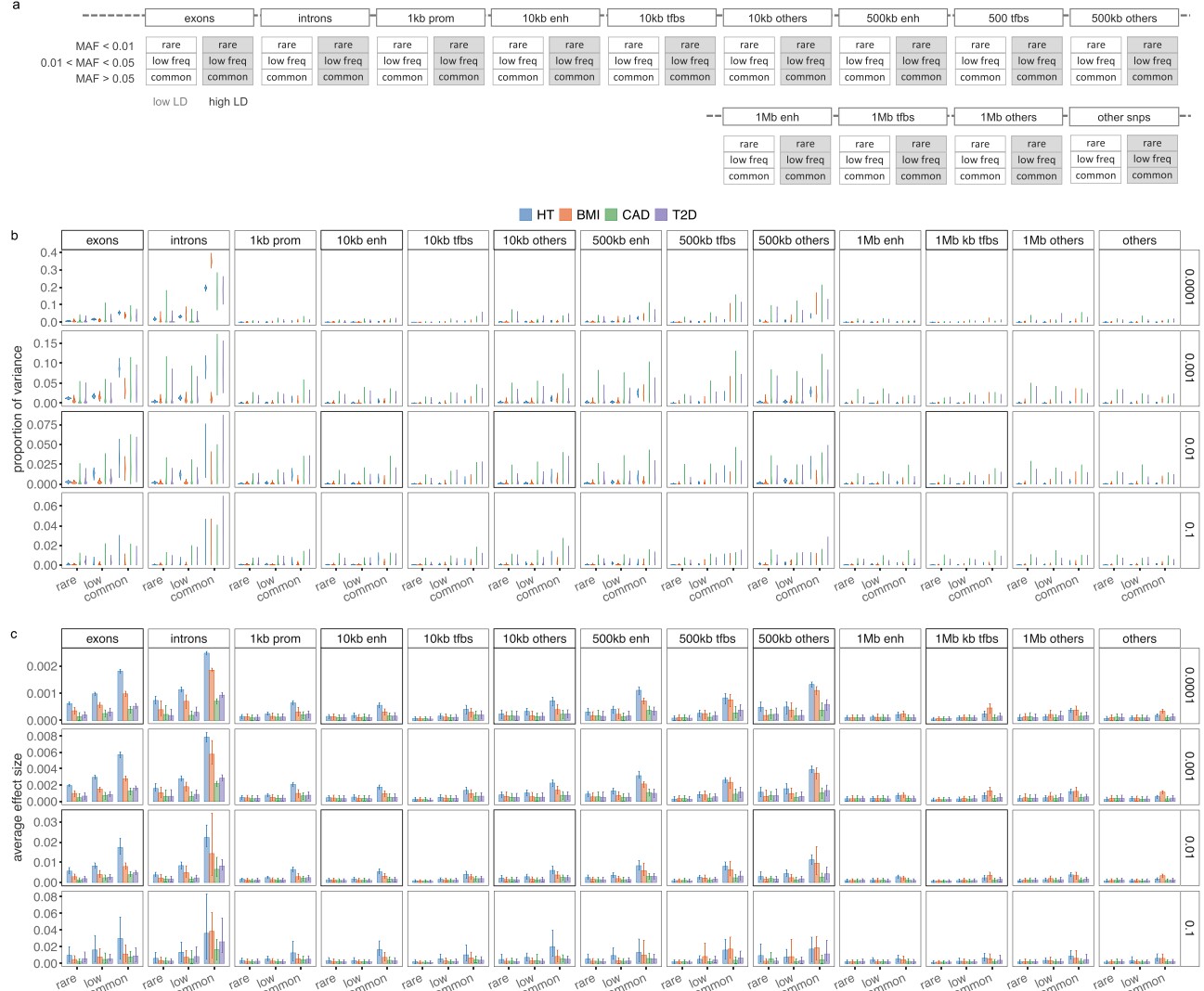

**Fig. 3 Genetic architecture of enrichment for height (HT), body mass index (BMI), cardiovascular disease (CAD) and type-2 diabetes (T2D) for 382,466 unrelated European ancestry UK Biobank individuals genotyped at 8,430,446 SNP markers. a** We partition SNP markers into seven location annotations (coding regions, intronic regions, and windows 1, 1–10, 10–500 kb and 500 kb–1 Mb upstream of genes, with other SNPs grouped in a category labelled "others"). Windows 1–10 kb, 10–500 kb and 500 kb–1 Mb upstream of genes are further split into SNPs mapped to enhancers (enh), transcription factor binding sites (tfbs) and others. Within each of the 13 annotations, we have three minor allele frequency groups (MAF ≤ 0.01 annotated as rare, 0.01 < MAF ≤ 0.05 annotated as low, and MAF > 0.05 annotated as common), and then each MAF group is further split into two based on median LD score. This gives 78 groups for which our BayesRR-RC model jointly estimates the phenotypic variation attributable to, and the SNP marker effects within, each group. For each of the 78 groups, SNPs were modelled using five mixture groups with variance equal to the phenotypic variance attributable to the group multiplied by constants (mixture 0 = 0, mixture 1 = 0.0001, 2 = 0.001, 3 = 0.01, 4 = 0.1). **b** Posterior distribution of the proportion of the total phenotypic variance attributable to the SNP markers that is contributed by each of the four non-zero mixtures within each MAF-annotation group for HT, BMI, CAD and T2D. Within these, are boxplots of the posterior mean and 95% credible intervals. Values are summed over LD groups. **c** Bar plots with error bars giving the 95% credible intervals for the average effect size of markers in the model for each MAF-annotation group, split by mixture.

a specific category, with perhaps the exception of high MAF variants (Fig. 3c). Generally, all phenotypes simply appear to be predominantly underlain by very many common variants, with SNPs within distal regulatory regions, coding and intronic regions contributing more to the variance. We also re-scaled the marker effects by the standard deviation of each marker, to give effect sizes on the allele substitution effect size scale, and again we find that rare variants have higher average allele substitution effects than common variants for exonic, intronic, promotors and enhancers (Supplementary Fig. 12b). An exception to these patterns were BMI-associated intronic and 10–500 kb group SNPs, where we find no evidence that the allele substitution effect size differs across frequency groups (Supplementary Fig. 12b). We

also did not find evidence that the allele substitution effect size differed across frequency groups for transcription factor binding sites, distal SNPs 1 MB upstream of genes, or those not mapping to an annotation group (Supplementary Fig. 12b).

**Discovery of associated genomic regions.** We then partitioned the variance attributed to SNP markers across 50kb regions of the genome, then across SNPs annotated to genes, and then to LD blocks of the DNA using our PPWV approach. We find 1660 50 kb regions for height with ≥95% posterior probability of explaining 0.001% of the $h^2_{SNP}$, 520 regions for BMI, 70 regions for CAD and 87 regions for T2D (Fig. 4a and Table 2). We then map

**Table 1 Proportion of genetic variance attributable to different genomic regions for height (HT), body mass index (BMI), type-2 diabetes (T2D) and cardiovascular disease (CAD).**

| Group | Trait | BayesRR-RC Posterior mean (95% CI) | RHE-mc[a] $h^2_{obs}$ (se) % | sLDSC[a] $h^2_{obs}$ (se) % | SumHer[a] $h^2_{obs}$ (se) % |
|---|---|---|---|---|---|
| Variance attributable to SNP markers genome-wide | HT | 57.66 (56.09, 59.14) | 63.28 (3.57) | 64.16 (2.86) | 98.58 (0.69) |
| | BMI | 28.74 (27.62, 30.0) | 26.76 (1.06) | 31.03 (0.9) | 44.98 (0.53) |
| | CAD | 5.94 (5.30, 6.67) | 4.49 (>100) | 4.73 (0.28) | 7.33 (0.43) |
| | T2D | 8.45 (7.83, 9.18) | 6.90 (0.47) | 6.53 (0.3) | 11.65 (0.44) |
| Proportion of genetic variance attributable to exonic regions of genes | HT | 24.75 (23.39, 26.071) | 27.09 | 3.00 | 16.74 |
| | BMI | 12.98 (10.98, 14.84) | 12.62 | 4.37 | 7.60 |
| | CAD | 13.23 (8.40, 18.84) | 18.68 | 1.69 | 15.34 |
| | T2D | 14.49 (10.74, 18.54) | 14.60 | 2.46 | 10.12 |
| Proportion of genetic variance attributable to intronic regions of genes | HT | 41.54 (39.91, 43.39) | 41.60 | 46.07 | 43.03 |
| | BMI | 44.17 (41.36, 47.25) | 47.87 | 44.61 | 48.19 |
| | CAD | 32.05 (24.98, 39.51) | 41.15 | 47.22 | 41.94 |
| | T2D | 37.28 (32.22, 42.57) | 48.66 | 38.52 | 48.02 |
| Proportion of genetic variance attributable to snps 1 kb upstream of genes | HT | 2.81 (2.24, 3.42) | 1.76 | 1.46 | 1.74 |
| | BMI | 1.62 (0.75, 2.69) | 0.36 | 1.90 | 1.15 |
| | CAD | 4.20 (1.71, 7.55) | 2.49 | <0.00 | 1.26 |
| | T2D | 3.58 (1.77, 5.86) | 3.40 | <0.00 | 1.57 |
| Proportion of genetic variance attributable to snps 10 kb upstream of genes | HT | 6.60 (5.84, 7.40) | 6.73 | 4.29 | 12.87 |
| | BMI | 5.28 (3.92, 6.87) | 3.19 | 6.58 | 4.10 |
| | CAD | 13.06 (8.70, 18.16) | 5.70 | 6.02 | 8.91 |
| | T2D | 9.08 (5.90, 13.28) | 4.02 | 20.44 | 7.56 |
| Proportion of genetic variance attributable to snps 500 kb upstream of genes | HT | 22.13 (21.00, 23.40) | 21.53 | 37.23 | 24.14 |
| | BMI | 28.58 (26.41, 31.01) | 28.81 | 35.86 | 31.17 |
| | CAD | 28.02 (21.24, 35.04) | 30.23 | 38.90 | 29.58 |
| | T2D | 27.42 (22.68, 32.36) | 24.33 | 32.49 | 27.47 |
| Proportion of genetic variance attributable to exonic regions that is explained by common variants | HT | 72.09 (69.77, 74.14) | 62.62 | 75.35 | 51.22 |
| | BMI | 69.41 (62.60, 76.42) | 59.67 | 16.43 | 54.31 |
| | CAD | 64.97 (43.08, 83.16) | 61.72 | >100 | 49.17 |
| | T2D | 68.57 (56.00, 79.82) | 66.33 | >100 | 64.11 |
| Proportion of genetic variance attributable to intronic regions that is explained by common variants | HT | 81.19 (79.30, 83.02) | 79.96 | 70.88 | 66.12 |
| | BMI | 85.05 (78.28, 91.49) | 86.10 | 70.62 | 69.68 |
| | CAD | 84.68 (65.64, 95.91) | 96.55 | 61.11 | 78.17 |
| | T2D | 87.62 (75.65, 94.85) | 87.63 | 67.93 | 71.39 |
| Proportion of genetic variance attributable to snps 500 kb upstream of genes that is explained by common variants | HT | 81.59 (78.91, 83.96) | 80.66 | 71.86 | 77.28 |
| | BMI | 86.78 (80.56, 91.60) | 89.95 | 67.38 | 74.81 |
| | CAD | 66.49 (49.11, 81.79) | 88.51 | 60.52 | 79.91 |
| | T2D | 72.35 (58.71, 83.75) | 94.91 | 69.48 | 75.12 |

[a]RHEmc[18], LDSC[19] and SumHer[6] provide the total SNP heritability observed (%) and single heritability estimates per genetic component (see Supplementary Data 2–5) that we summarised to obtain the proportion of genetic variance attributed to exonic regions, intronic regions and windows 1, 1–10 and 10–500 kb upstream of genes.

SNPs to their closest gene ($+/-50$ kb from SNP position) and we use our annotations to label them (see "Methods" section). We find 243 independent coding regions for height with ≥95% posterior probability of explaining at least 0.001% of the $h^2_{SNP}$, 29 independent coding regions for BMI, 5 for CAD and 13 for T2D. We find many more associations in the cis region of genes with 1254 independent cis-regions for height with ≥95% posterior probability of explaining 0.001% of the $h^2_{SNP}$, 1765 independent cis-regions for BMI, 1166 for CAD and 1221 for T2D. We additionally find 9 independent promoter regions and 1072 independent introns for height with ≥95% posterior probability of explaining at least 0.001% of the $h^2_{SNP}$, 1162 independent intronic gene regions for BMI, 307 for CAD and 347 for T2D. When we

calculate the number of exons, introns, promotors and cis regions with ≥95% posterior probability of explaining 0.001% of the $h^2_{SNP}$, as a proportion of the total number within each chromosome, we find that up to 24% of the genes on each chromosome are associated with each of the four traits (Fig. 4b). Generally, we find that only 1% or less of the available exons and promotor regions of genes per chromosome show an association with each of the phenotypes, but up to 14% of the available intronic regions and up to 10% of the cis-regions surrounding genes contribute to the phenotypic variance with ≥95% probability (Fig. 4b). The variance contributed by each exonic, intronic, promotor, or cis region is typically only a small fraction of a percent, with largest effect sizes being the exonic region of GDF5 contributing 0.26%

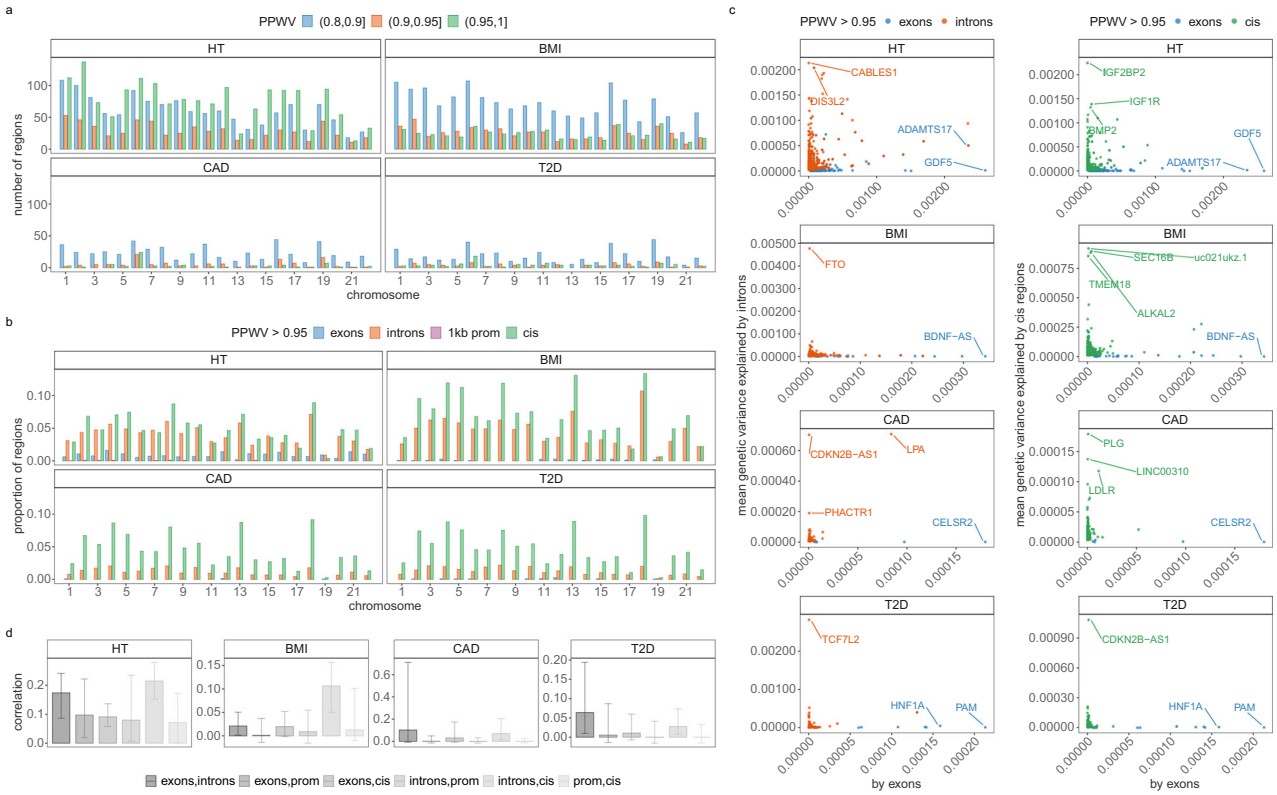

**Fig. 4 Contribution of genes and 50kb regions to height (HT), body-mass-index (BMI), cardiovascular disease (CAD) and type-2-diabetes (T2D).**
**a** We grouped SNPs in 50 kb-regions genome-wide and estimated the sum of the squared regression coefficient estimates for each 50 kb-region. We then select the number of 50 kb regions that explain at least 0.001% of the variance attributed to all SNP markers in 80, 90 and 95% of the iterations. This gives a measure called the posterior probability that the window variance (PPWV)[20] exceeds 1/10,000 of the phenotypic variation attributed to SNP markers. **b** We mapped SNPs to the closest gene +/−50 kb from the SNP position and labelled them as located in a coding region, an intron, 1 kb upstream of a gene using our functional annotations (Fig. 3a). Remaining snps are labelled as located in a cis-region (up to +/−50 kb from a gene). We then select the number of regions where PPWV is higher than 95% and explains at least 0.001 % of the phenotypic variance attributed to all SNP markers. We then calculate the number of significant coding regions, introns, 1 kb regions and cis regions as a proportion of the total number of genes for each chromosome. Genic associations that explain at least 0.001% of the phenotypic variance attributed to all SNP markers are again spread across chromosomes according to the chromosome length. **c** Shows the mean of the phenotypic variance attributed to intron and cis regions (y-axis) and coding regions (x-axis) that explain at least 0.001% of the phenotypic variance attributable to SNP markers in ≥95% of the iterations (PPWV > 0.95). These results provide joint estimates of the proportions of variance contributed by different gene bodies and automatic fine-mapping of gene bodies and their cis-regulatory regions. For example, introns and cis-regulatory regions of FTO respectively contribute 0.48% (95% CI 0.29, 1.12) and 0.01% (95% CI 0, 0.01) to the phenotypic variance of BMI. **d** We calculated the phenotypic variance contributed by exonic, intronic, promoter region and SNPs +/−50 kb outside of the exon and promotor regions (cis) for each gene. Bar plots show the correlation among the variance explained by the groups across genes. Error bars show the SD.

(95% CI 0.21, 0.32) to the phenotypic variance of height, the intronic region of FTO contributing 0.48% (95% CI 0.29, 1.12) to BMI, both the exonic-region and intronic-region of LPA contributing a combined 0.08% (95% CI 0.04, 0.13) to the risk of CAD, and the intronic region of TCF7L2 contributing 0.28% (95% CI 0.23, 0.35) to the risk of T2D (Fig. 4c, full results in Supplementary Data 6–9). Taken together, these results support an infinitesimal contribution of many thousands of genes to common complex trait variation and give joint estimates of the proportions of variance contributed by each gene and their probability of association.

For each gene, we also calculated the phenotypic variance contributed by exonic, intronic, promotor region, and cis SNPs and then calculated the correlation among the variances explained by the groups across genes. Across traits, we find small positive correlations of the variance attributable to exonic and intronic regions of 0.17 (0.09, 0.24 95% CI) for height, 0.02 (0.001, 0.05 95% CI) for BMI, 0.103 (−0.007, 0.71 95% CI) for CAD, and 0.064 (0.01, 0.19 95% CI) for T2D. Similarly, we find small positive correlations between introns and cis regions

(Fig. 4d). With the exception of height, there was no evidence for a relationship among the following groups: (i) SNPs in the exons of each gene and SNPs +/−50 kb outside of the exon and promotor regions; (ii) SNPs in the exons of each gene and SNPs in proximal promotors; and (iii) intronic SNPs and SNPs in promotor regions (Fig. 4d). This implies that trait associated SNPs in proximal and distal regulatory regions are largely independent of the effects of SNPs in their closest exon, as they do not align in terms of the variance they explain (Fig. 4d). For height, small weakly positive correlations across all gene regions in their contribution to variance, implies a degree of alignment across genes in regulatory variants and the closest exon (Fig. 4d). These results suggest a regulatory link between introns and distal cis regions outside of the promotor, or that introns may be correlated with structural variation. They also imply that the variance contributed by regulatory regions and those in the closest coding regions are not strongly coupled for these common complex traits.

Finally, our approach provides automatic fine-mapping of SNP loci, and of these region-level and gene-level associations, 360

**Table 2 Summary of findings for height (HT), body mass index (BMI), type-2 diabetes (T2D) and cardiovascular disease (CAD).**

| Findings | Method | HT | BMI | CAD | T2D |
|---|---|---|---|---|---|
| Associated SNPs | COJO-plink2 | 1673 | 517 | 34 | 85 |
| | COJO-BoltLMM | 2131 | 565 | 34 | 84 |
| | COJO-Regenie | 2134 | 555 | 34 | 82 |
| 50 kb regions (PPWV ≥ 95%) | BayesRR-RC | 1660 | 520 | 70 | 87 |
| Genic regions (PPWV ≥ 95%) | BayesRR-RC | 2578 | 2956 | 1478 | 1581 |
| Exons | | 243 | 29 | 5 | 13 |
| Introns | | 1072 | 1162 | 307 | 347 |
| cis[a] | | 1254 | 1765 | 1166 | 1221 |
| SNPs (PIP ≥ 95%) | BayesRR-RC | 360 | 20 | 2 | 9 |
| Exons | | 216 | 16 | 1 | 4 |
| Introns | | 73 | 2 | 1 | 5 |
| 10–500 kb | | 48 | 1 | 0 | 0 |
| LD clumps with $r2 = 0.1$ (PPWV ≥ 95%) | BayesRR-RC | 1220 | 206 | 16 | 19 |

[a]SNPs located up to +/−50 kb from the closest gene.

SNPs for height, 20 for BMI, 2 for CAD and 9 for T2D could be mapped to a single SNP with greater than 95% inclusion probability across all four chains (Supplementary Data 10 and Supplementary Fig. 13). Of these fine-mapped SNPs, only 53.45% are top loci with a p-value $< 5 \times 10^{-8}$ from the fastGWAS UK Biobank summary statistic data for standing height, BMI, angina/heart attack and type-2 diabetes (fastGWA, see "Code availability"). This highlights that selecting on the top SNP markers identified by standard association studies would give a different set of variants than those obtained from selecting high PIP SNPs.

**Out-of-sample prediction into another European healthcare system.** We generated a full posterior predictive distribution for each trait in each of 32,500 individuals from the Estonian Genome Centre data, which allows the transmission of uncertainty in the marker effect estimates from the UK Biobank to the genomic predictors created in Estonia. First, despite this study having almost half the sample size, we show improved genomic prediction as compared to recently proposed summary statistic approaches[23], when taking the mean of the predictor across iterations and correlating this with the phenotype with correlation of 0.62 for height, 0.34 for BMI, 0.16 for T2D, and 0.07 for CAD (Supplementary Fig. 14a). The area under the receiver operator curve (AUC) for T2D was 0.67 and 0.57 for CAD. In comparison, using the 64 BLD-LDAK annotations recommended by a recent study[21], the highest prediction accuracy obtained from MegaPRS was 0.55 for height, 0.32 for BMI, 0.10 for T2D, and 0.05 for CAD.

We then estimated the distribution of the partial correlations between the trait and genomic predictors created from our different annotation groups and find that exonic, intronic, and 10–500 kb upstream regions contribute proportionally more to the prediction accuracy than other genomic groups, replicating our results from the UK Biobank (Supplementary Fig. 14). We find evidence for zero/low correlations of genomic predictors created from different annotation groups, which supports our results from the UK Biobank (Supplementary Fig. 14e). This suggests that individuals have a different portfolio of risk variants, with different genomic regions contributing for different

individuals to their overall genetic value, as expected under a highly polygenic model.

Additionally, for height and BMI we also determined the proportion of the posterior predictive distribution for each individual that was within +/−1 SD of their true phenotypic value. On average 67.5% of an individuals posterior predictive distribution is within +/−1 SD of their true phenotype for BMI and 75% for height, with similar prediction accuracy across individuals (Supplementary Fig. 14c). For T2D and CAD, we extended the PCF metric, typically defined as the proportion of cases with larger estimated risk than the top pth percentile of the distribution of genetic risk in the general population. For each individual, we calculated the proportion of their posterior predictive distribution that falls above the top 25% of the distribution of genetic risk in the general population. The distribution of these probabilities is shown for confirmed cases and those without diagnosis in the Estonian Biobank (Supplementary Fig. 14d). We find 25 individuals for T2D and 15 individuals for CAD where ≥90% of their posterior predictive distribution is within the high risk group of which 40 and 18% are currently defined as cases for T2D and CAD, respectively based on recent medical records. This is compared to 1% and 2% case rate for those with ≤10% probability of being in the high risk group for T2D and CAD respectively, giving an odds ratio of 20 and 18 between the ≥90% and ≤10% groups. However, our results clearly show that the individual-level sensitivity and specificity of genomic prediction for these common complex diseases is very poor, as 75% of T2D cases and 92% of CAD cases have ≤50% of their distribution within the high-risk category. These results highlight how variation contained within a posterior predictive distribution that is typically ignored in human genomic prediction can be used. We show that genomic prediction for personalised medicine with patient-specific predictions or stratification of patients is currently extremely limited.

## Discussion

There is no single statistical model appropriate for all settings and thus there will always be a situation where a model poorly fits the data. We have provided theoretical and empirical evidence that a grouped Dirac spike-and-slab model (which we term BayesRR-RC), has a prior that is flexible enough to show robust model performance across the data analysed here, improving inference in many settings over commonly applied approaches. We develop a range of computational and statistical approaches which allow this, or any similar Gibbs sampling algorithm, to scale to whole genome sequence data on many hundreds of thousands of individuals. This has enabled us to compare and contrast the inferred underlying genetic distribution for four complex phenotypes under this prior, providing novel insight into the genetic architecture of these traits. We observe that all phenotypes simply appear to be predominantly underlain by very many common variants, with SNPs within distal regulatory regions, coding and intronic regions each contributing more to the phenotypic variance and having higher allele substitution effects.

There has been debate on how to best estimate SNP heritability[1,3,4] and here we validate that one approach could be to split SNP markers by LD to improve genetic effect size estimates. Our results suggest that the proportion of genomic variation attributable to mutations in regulatory regions and mutations in the closest genic regions are largely independent. Additionally our model tests association within groups in a probabilistic way and we find 290 independent coding, 2888 independent intronic, and 5406 independent cis regions with ≥95% probability of contributing at least 0.001% of the SNP heritability. Understand how these coding, intronic and proximal and distal regulatory regions combine to contribute to

phenotypic variance remains a substantial challenge and our results suggest a predominant role for introns and for distal, and thus likely more global enhancers, rather than locally dominant proximal expression QTL. The recent "omnigenic" model[24], suggests that trait-associated variants in regulatory regions influence a local gene which is not directly causal to the disease, and also co-regulate other disease causal genes (or "core" gene). Our findings of little correlation of exonic and proximal regulatory variance and a large number of trait-associated intronic and cis regions do not rule this out, but suggest a more complex infinitesimal picture with differences occurring among traits, potentially due to their evolutionary history.

There are important caveats and limitations to consider. Here, we present an approach for analysing large-scale biobank data, which is becoming increasingly available, However, a substantial number of GWAS have already been conducted, with associated published genome-wide summary association statistic estimates. Many methods have been developed to take advantage of these estimates, with downstream analysis models making use of various summary statistics resources in efficient and flexible ways. We show here that two leading summary statistic approaches perform poorly as compared to individual-level models for estimation of enrichment and genomic prediction. Despite this, the sample sizes obtained in consortia study meta-analyses will exceed those from single biobanks, especially for disease, and thus the genomic prediction accuracy of consortia study meta-analysis summary statistic prediction models may exceed those from individual-level analyses. Combining the posterior distribution obtained from BayesRR-RC across different individual-level biobank studies would alleviate this issue.

Additionally, in this work we do not extend past a limited number of functional annotations and thus we do not provide a model capable of further partitioning the variation into specific regulatory functions (eQTL, mQTL, pQTL etc.) or directly modelling the relationships among components. LDSC functional methods take the approach that SNPs can be assigned to different categories (e.g., both coding and conserved), with the categories competing against each other to explain the signal, with the downside that enrichment is relative and that the total variance is not partitioned. Here, the total variance is partitioned but this is based on preferential allocation of SNPs to coding regions, then introns, and then to their nearest upstream gene position. These SNPs are most likely to be allocated accurately, with 1 and 1–10 kb groups being more ambiguous in high gene density regions and likely mislabelled. However, if this was the case then variance would still be partitioned to these mislabelled groups and it would just be evenly split across them, with experimentally validated promotor, enhancer and tfbs regions assisting to some degree in alleviating this. Rather, here we see a clear pattern of increasing variance contributed, increasing average effect size, and an increasing pattern of higher rare allele substitution effects by individual markers as distance from the nearest gene increases. 10–500 kb distal regions may contribute more variance as marker density and marker coverage is higher in these regions, with missing variation within 10 kb upstream as causal variants are poorly correlated with SNPs. The posterior distributions for the variance explained by 1 kb, 1–10 kb regions, and 10–500 kb regions are negatively correlated (Supplementary Fig. 8, meaning that these groups are competing with each other, as if variance goes to one then it is being taken away from the other because they are in LD), and thus there is the risk that the model cannot separate these effectively. However, this is true of any enrichment analysis conducted to date and we can only make inference in the data that we have currently available. Resolving this requires the application of this model to whole genome sequence data where the total variance can be partitioned across upstream regions without marker coverage concerns. Irrespective of exactly which

upstream region variance is allocated to, our inference that genic regions are uncorrelated in their contribution to variance with the promotor and upstream regions still holds as does our probabilistic inference on the associations of each gene and their contribution to the phenotypic variation.

Our results provide evidence for an infinitesimal contribution of many thousands of common genomic regions to common complex trait variation and for a predominant role of intronic, exonic, and distal regulatory regions. This highlights the immense challenge of understanding the molecular underpinning of each association and the difficulties in improving the estimation of many tens of thousands of small-effect associations that are required to improve genomic prediction. This work represents a step toward maximising the probabilistic inference that can be obtained from large-scale Biobank studies.

## Methods

**BayesRR-RC model**. We extend the BayesR model to a BayesRR-RC model as follows

$$\mathbf{y} = \mathbf{1}\mu + \sum_{\varphi=1}^{\Phi} \mathbf{X}_\varphi \boldsymbol{\beta}_\varphi + \boldsymbol{\epsilon}, \tag{3}$$

where there is a single intercept term $\mathbf{1}\mu$ and a single error term $\boldsymbol{\epsilon}$ but now SNPs are allocated into groups $(\varphi_1, \ldots, \varphi_\Phi)$, each of which having it's own set of model parameters $\Theta_\varphi = \{\beta_\varphi, \pi_{\beta_\varphi}, \sigma_{G_\varphi}^2\}$. As such, each $\beta_{\varphi_j}$ is distributed according to:

$$\beta_{\varphi_j} \sim \pi_{0_\varphi}\delta_0 + \pi_{1_\varphi}\mathcal{N}\left(0, \sigma_{1_\varphi}^2\right) + \pi_{2_\varphi}\mathcal{N}\left(0, \sigma_{2_\varphi}^2\right) + \ldots + \pi_{L_\varphi}\mathcal{N}\left(0, \sigma_{L_\varphi}^2\right), \tag{4}$$

where for each SNP marker group $\{\pi_{0_\varphi}, \pi_{1_\varphi}, \ldots, \pi_{L_\varphi}\}$ are the mixture proportions and $\{\sigma_{1_\varphi}^2, \sigma_{2_\varphi}^2, \ldots, \sigma_{L_\varphi}^2\}$ are the mixture-specific variances prop ortional to

$$\begin{bmatrix} \sigma_{1_\varphi}^2 \\ \vdots \\ \sigma_{L_\varphi}^2 \end{bmatrix} = \sigma_{\beta_\varphi}^2 \begin{bmatrix} C_{1_\varphi} \\ \vdots \\ C_{L_\varphi} \end{bmatrix}$$

Thus the mixture proportions, variance explained by the SNP markers, and mixture constants are all unique and independent across SNP marker groups. This extends previous models (known as BayesRC[25] and BayesRS[26]), which have used additional mixtures for different SNP groups, but kept a single global variance component. Importantly, a single variance component with more mixtures serves only to change the amount of mass allocated at different sizes of the distribution, but does not alter the sizes of the effects themselves as there is still a single distribution. In contrast, the formulation presented here of having an independent variance parameter $\sigma_{\beta_\varphi}^2$ per group of markers, and independent mixture variance components, enables estimation of the amount of phenotypic variance attributable to the group-specific effects and enables differences in the distribution of effects among groups. In this work, we use 78 SNP marker groups, each with five mixture components (including 0).

We can sketch the difference in the models by looking at the respective conditional posteriors, again, assuming a single component for simplification purposes. We have a BayesRC or BayesRS estimator by assuming different groups of effects as described in Supplementary Note 4 Eq. 35, which yields:

$$f\left(\alpha, \gamma | \pi_{\beta_\varphi}, \sigma_\beta^2, \sigma_\epsilon^2, y\right) \propto \exp\left\{\frac{1}{2\sigma_\epsilon^2}||y - \mathbf{X}_{\gamma\neq0}\alpha_{\gamma\neq0}||_2^2 - \frac{1}{2\sigma_\beta^2}||\alpha||_2^2 - \log\left(\frac{1-\pi_{\beta_\varphi}}{\pi_{\beta_\varphi}}\right)||\gamma_\varphi||_0\right\}, \tag{5}$$

where $\pi_{\beta_\varphi}$ are the group-specific mixture proportions and $||\gamma_\varphi||_0$ is the cardinality of the group. The corresponding MAP estimate would amount to adding extra penalisation on sparsity through the $\pi_\varphi$ terms, while keeping the same level of shrinkage as the baseline BayesR.

In our model the conditional posterior is:

$$f\left(\alpha, \gamma | \pi_{\beta_\varphi}, \sigma_{\beta_\varphi}^2, \sigma_\epsilon^2, y\right) \propto \exp\left\{\frac{1}{2\sigma_\epsilon^2}||y - \mathbf{X}_{\gamma\neq0}\alpha_{\gamma\neq0}||_2^2 - \frac{1}{2\sigma_{\beta_\varphi}^2}||\alpha||_2^2 - \log\left(\frac{1-\pi_{\beta_\varphi}}{\pi_{\beta_\varphi}}\right)||\gamma_\varphi||_0\right\} \tag{6}$$

now each marker has a group-specific shrinkage $\sigma_{\beta_\varphi}^2$, which translates to a specific $\lambda_\varphi$ per group in the MAP estimate. This amounts to markers being shrunk according to the scale of the effects of their group, instead of the scale of all other markers. So instead of solving a single model selection and regularisation problem we are solving $\Phi$ model selection and regularisation problems, with shared information only through the residuals. If we subset by MAF and LD bins, the resulting groups of columns will have a correlation pattern similar to an exponential decay (LD decays with distance). If we take the whole genotype matrix, the pattern would be closer to a block diagonal matrix of correlations, in refs. [16,27]

it is showed that the former case requires weaker conditions in order to recover the true vector $\beta$ consistently than the latter. Although the sampling scheme was different, we have shown that a similar model with only two groups: genetic markers and epigenetic markers, is successful in identifying BMI and smoking epigenetic signatures[13]. The baseline model derivations for this model are outlined in Supplementary Note 1, a BSP Gibbs sampling scheme and an assessment of its performance is outlined in Supplementary Note 2, and an assessment of the model performance with correlated covariates is outlined in Supplementary Note 4.

## Simulation study

*Genetic architecture.* We first compare the model performance of BayesRR-RC to existing approaches across 18 different genetic architectures. We randomly selected 40,000 unrelated UK Biobank individuals and used 596,741 imputed SNP markers from chromosomes 19 to 22. We randomly selected either 1000, 10,000, or 100,000 LD independent (LD $R^2 < 0.1$) causal SNP markers. For each SNP marker set there were two settings.

In the first setting, we simulated effect sizes from a normal distribution with zero mean and variance of 0.1, 0.3, or 0.6 divided by the number of causal variants $\propto N(0, [p(1-p)]^{-0.25})$, with $p$ the allele frequency. We sampled individual-level environmental (residual) variance from a normal distribution with zero mean and variance equal to 1 minus either 0.1, 0.3, or 0.6 to give phenotypes with zero mean and unit variance. This gave $h^2_{SNP} = 0.1$, 0.3, or 0.6 and simulates stronger effect sizes for rare variants in line with recent empirical estimates. We simulated ten replicate phenotypes for each of the nine different genetic architectures. In the second setting, we repeat each simulation, sampling the SNP marker effects from 13 different normal distributions, one for each of 13 different genomic annotation groups described in the main text. The 13 groups were allocated different proportions of the $h^2_{SNP}$ as follows: for exonic variants $P(h^2_{SNP}) = 0.167$, intronic variants $P(h^2_{SNP}) = 0.334$, 1 kb promotor variants $P(h^2_{SNP}) = 0.0835$, 1–10 kb enhancer variants $P(h^2_{SNP}) = 0.04175$, 1–10 kb transcription factor binding sites $P(h^2_{SNP}) = 0.04175$, 1–10 kb other variants $P(h^2_{SNP}) = 0$, 10–500 kb enhancers $P(h^2_{SNP}) = 0.0835$, 10–500 kb transcription factor binding sites $P(h^2_{SNP}) = 0.0835$, 10–500 kb other variants $P(h^2_{SNP}) = 0$, 500 kb–1 Mb enhancers $P(h^2_{SNP}) = 0.0835$, 500 kb–1 Mb transcription factor binding sites $P(h^2_{SNP}) = 0.0835$, 500 kb–1 Mb other variants $P(h^2_{SNP}) = 0$, and other non-annotated SNPs $P(h^2_{SNP}) = 0$. For each of the 13 groups marker effects were simulated as $\propto N(0, [p(1-p)]^{-0.25})$ to give $h^2_{SNP} = 0.1$, 0.3, or 0.6, with stronger effect sizes for rare variants. Four of these 13 groups had zero variance indicating that no associations were created for these groups.

Thus, in the first setting we simulate variance explained by annotation groups that is on average proportional to the number of SNPs within each annotation (due to the random allocation of SNPs and effect sizes). In the second setting, the variance and average effect size differ across annotation groups. We refer to these as two different enrichment settings: "random", or "enriched".

For these 180 phenotypes, we ran the following individual-level models:

- A restricted maximum likelihood model implemented in the software GCTA with a single relationship matrix providing an estimate of the variance attributable to SNPs genome-wide.
- A restricted maximum likelihood model implemented in the software BoltREML[17]. Here, we used a 78 MAF-LD-annotation group model using the non-overlapping genomic annotation groups described below in the UK Biobank analysis providing an estimate of the variance attributable to SNPs genome-wide and an estimate of the variance attributable to SNP markers of each annotation group.
- A Haseman-Elston regression using the same 78 group model implemented in the software RHEmc[18], providing an estimate of the variance attributable to SNPs genome-wide and an estimate of the variance attributable to SNP markers of each annotation group.
- Mixed linear association model (MLMA), which is a two-stage approach where the variance attributable to the SNP markers genome-wide is estimated and this estimate is then used in a second generalised least squares step to test for SNP-phenotype associations one marker at a time. There are two forms of this model. In the first, the SNP is fitted twice as it is included in both the fixed and random terms (MLMAi). In the second, the SNP to be tested as fixed is removed from the random term alongside those on the same chromosome (MLMA). We used the software BoltLMM[8], Regenie[9], and GCTA to fit these models. These approaches provided estimates of the SNP regression coefficients (marker effect sizes).
- Single marker marginal least squares regression using plink2[28], whilst fitting 20 principal components of the marker data as covariates.
- Linkage disequilibrium score regression (LDSC[19]), with LD scores calculated using the same data, and the same 78 non-overlapping annotations in a 78 component LDSC annotation model. We included SNPs with MAF > 1% following the software instructions. This model is intended to approximate an individual-level REML analysis with 78 annotations and provides an estimate of the variance attributable to SNPs genome-wide and an estimate of the variance attributable to SNP markers of each annotation group.
- We used the software SumHer[6]. We calculated marker taggings under the same 78 component annotation model. We ignored the LD weights when

calculating the taggings as we found this gave the best estimates we could obtain from the simulated data across all scenarios. We set the relationship of effect size and minor allele frequency to be $-0.25$ as suggested by the authors and which matches the simulation setting. This model is intended to approximate an individual-level REML analysis with 78 annotations, but using a different scaling of the relationship matrix, and provides an estimate of the variance attributable to SNPs genome-wide and an estimate of the variance attributable to SNP markers of each annotation group.

- Our BayesRR-RC model implemented in GMRM with 78 SNP-marker groups and run for 5000 iterations with a burn-in period of 2000 iterations.
- Our BayesRR-RC model implemented in GMRM with only a single SNP-marker group, which is equivalent to BayesR, run for 5000 iterations with a burn-in period of 2000 iterations.

We then ran the following prediction models, using a testing set of 10,000 UK Biobank unrelated individuals, that were also unrelated to the training data, and focusing on the models proposed in a recent paper[21]. These methods contain two approximations to our BayesRR-RC model and the authors claim to outperform all other existing methods, including individual-level models. The models are:

- An individual-level bayesR model using genomic annotation SNP variance estimates from the SumHer models as implemented in the software MegaPRS[21]. This provides estimates of the SNP marker effects for creating a genetic risk predictor.
- An individual-level boltREML model using genomic annotation SNP variance estimates from the SumHer models as implemented in the software MegaPRS[21]. This provides estimates of the SNP marker effects for creating a genetic risk predictor.
- A summary statistic bayesR model using genomic annotation SNP variance estimates from the SumHer models as implemented in the software MegaPRS[21]. This provides estimates of the SNP marker effects for creating a genetic risk predictor.
- A summary statistic boltREML model using genomic annotation SNP variance estimates from the SumHer models as implemented in the software MegaPRS[21]. This provides estimates of the SNP marker effects for creating a genetic risk predictor.

First, we compared the correlation of the simulated and estimated proportion of phenotypic variance attributable to the 13 genomic annotation groups across all models in Fig. 1. We determined the ability of the approaches to correctly identify enriched regions of the DNA by estimating the probability within each simulation replicate that a SNP marker group would have an estimated enrichment of ≥2 (i.e., being described as having average effect sizes that are twice as large as expected) when the simulated value was ≤1.1. As BayesRR-RC induces sparsity in the SNP effect estimates, with some markers always remaining in the variance = 0 spike, we propose a different enrichment definition where the proportion of $h^2_{SNP}$ is divided by the proportion of markers that are in the model for the SNP group, rather than the proportion of markers mapping to the SNP group.

In Supplementary Note 3, we propose a posterior probability window variance (PPWV) approach[20], which provides a probabilistic determination of association of a given LD block, genomic window, gene, or upstream region, relative to the amount of phenotypic variation attributable to that window. Our PPWV approach determines the posterior inclusion probability that each region and each gene contributes at least 0.001% to the $h^2_{SNP}$, with theory and small-scale simulations outlined in Supplementary Note 3 suggesting well controlled FDR. We partitioned the 596,741 imputed SNP markers in LD blocks, defined as groups of markers with LD $R^2 \geq 0.1$. Within each simulation replicate, we estimated the probability that LD blocks containing a causal variant were identified by PPWV. We compared this to MLMA estimates obtained using the BoltLMM software, by estimating the probability that LD blocks containing a causal variant were identified as having a SNP with $p$-value $\leq 5 \times 10^{-8}$, the standard genome-wide significance threshold. We present these results in Fig. 2a.

We then compare the prediction accuracy obtained in a testing set of 10,000 UK Biobank unrelated individuals, that were also unrelated to the training data. We predicted phenotype using SNP marker effect sizes obtained from BayesRR-RC, MLMA effect sizes from BoltLMM, and the four MegaPRS methods outlined above implemented in the LDAK software. While we would suggest that fixed-effect MLMA estimates are improper for prediction we include this comparison as polygenic risk scores have often been created from fixed-effect SNP estimates. We calculate the correlation between the simulated phenotype in the testing set and the genomic predictor within each simulation replicate and we compare the mean correlation across the 18 different genomic annotations in Fig. 2. Additionally, to provide a benchmark, we compare to the theoretical expectation under ridge regression approximations[29], with the number of markers set to the number of causal variants.

*Relationship between effect size, minor allele frequency and LD.* We then conducted another large-scale, but this time well-powered simulation study, where we ascertained the causal variant SNP markers in different ways and varied the relationship between effect size, minor allele frequency and LD. We used the same randomly selected 40,000 unrelated individuals and all 596,741 imputed (version 3) genetic markers from chromosomes 19 through 22 from the UK Biobank. We

simulated a wide-range of different possible underlying genetic effect size distributions as follows:

- We chose either 5000 or 10,000 imputed SNP markers for which to assign a genetic effect size, providing two different levels of polygenicity.
- We selected these 5000 or 10,000 markers in two different ways. Either, we selected SNPs at random, or we selected the marker of highest minor allele frequency per LD block of the genome, with an LD block defined as a group of SNP markers with absolute LD of at least 0.05. Randomly allocating markers creates a set of associated variants with the same distribution of LD and MAF as the SNP data, which is composed of predominantly low frequency variants. Selecting only the highest frequency marker per LD block creates a setting where for each set of markers in LD with each other, there is only one causal genetic variant, and where the distribution of associated markers differs to that of the SNP markers as a whole.
- Having created four different ways of selecting associated markers (5000 or 10,000 and high-MAF or random) we then created five different ways of assigning effect sizes to them:

  – We simulated effect sizes from a normal distribution with zero mean and variance 0.6 divided by the number of markers (5000 or 10,000) with no relationship to the LD or MAF of the markers. Thus, effects had variance $\propto N(0, w^0[p(1-p)]^0)$ with $w$ the LD score of the marker and $p$ the allele frequency.
  – We simulated effect sizes from a normal distribution with zero mean and variance 0.6 divided by the number of markers (5000 or 10,000) $\propto N(0, w^{-0.25}[p(1-p)]^{-0.25})$. This simulates stronger effect sizes for rare variants and those in low LD.
  – We simulated effect sizes from a normal distribution with zero mean and variance 0.6 divided by the number of markers (5000 or 10,000) $\propto N(0, w^{0.25}[p(1-p)]^{-0.25})$. This simulates stronger effect sizes for rare variants and those in high LD.
  – We simulated effect sizes from a normal distribution with zero mean and variance 0.6 divided by the number of markers (5000 or 10,000) $\propto N(0, w^{-0.25}[p(1-p)]^{0.75})$. This simulates equivalent effect sizes for common and rare variants, and greater effects for markers in low LD.
  – We simulated effect sizes from a normal distribution with zero mean and variance 0.6 divided by the number of markers (5000 or 10,000) $\propto N(0, w^{0.25}[p(1-p)]^{0.75})$. This simulates equivalent effect sizes for common and rare variants, and greater effects for markers in high LD.

- For each of the four different sets of markers, each with five different effect size sampling schemes, we then created two additional settings. In the first setting markers were sampled from the various normal distribution, as described above, for the five different effect size sampling schemes. In the second setting, for each of the five different effect size sampling schemes we simulated effects from 13 different distributions, one for each of 13 different genomic annotation groups with different proportions of total SNP heritability ($h^2_{\mathrm{SNP}}$). For each of the five different effect size sampling schemes the relationship to LD and MAF remained the same, but the total variance attributed to the SNP markers was partitioned across annotation groups as follows for exonic variants ($h^2_{\mathrm{SNP}} = 0.1$), intronic variants ($h^2_{\mathrm{SNP}} = 0.2$), 1 kb promotor variants ($h^2_{\mathrm{SNP}} = 0.05$), 1–10 kb enhancer variants (0.025), 1–10 kb transcription factor binding sites ($h^2_{\mathrm{SNP}} = 0.025$), 1–10 kb other variants ($h^2_{\mathrm{SNP}} = 0$), 10–500 kb enhancers ($h^2_{\mathrm{SNP}} = 0.05$), 10–500 kb transcription factor binding sites ($h^2_{\mathrm{SNP}} = 0.05$), 10–500 kb other variants ($h^2_{\mathrm{SNP}} = 0$), 500 kb–1 Mb enhancers ($h^2_{\mathrm{SNP}} = 0.05$), 500 kb–1 Mb transcription factor binding sites ($h^2_{\mathrm{SNP}} = 0.05$), 500 kb-1 Mb other variants ($h^2_{\mathrm{SNP}} = 0$), and other non-annotated SNPs ($h^2_{\mathrm{SNP}} = 0$). Four of these distributions had zero variance indicating that no associations were created for these groups. In the first setting, this simulates variance explained by annotation groups that is on average proportional to the number of SNPs within each annotation. In the second scheme, the variance and average effect size differs across annotation groups. We refer to these as two different enrichment settings: "random", or "enriched".
- This created four different sets of associated markers (5000 or 10,000 and high-MAF or random), each with five different marker effect size sampling schemes, which we refer to in the main text as the 20 different generative genetic models (Table 1), each of which has two enrichment settings. This gave 40 different sampling schemes for the genetic effects and we simulated ten replicates for each setting, giving a total set of 400 simulated phenotypes.
- For each generative model the total genetic variance was 0.6 and we sampled individual-level environmental (residual) variance from a normal distribution with zero mean and variance 0.4 to give phenotypes with zero mean and unit variance.

This range covers generative genetic models discussed in the literature and provides models that both fit and violate the assumptions of the range of variance component statistical models. This includes both individual-level and summary statistic approaches, that are currently applied in the literature for estimation of the variance attributable to the SNP markers, for testing association of genetic markers with phenotypes genome-wide, and for genomic prediction.

This simulation provides a range of different scenarios for which we can explore the model performance of BayesRR-RC and compare it to existing approaches. In Supplementary Fig. 1, we compare the $h^2_{\mathrm{SNP}}$ estimation, estimation of the annotation genetic variance along with the RMSE of the estimates, and the estimated average effect size.

We then extend our model comparisons in a number of ways. While direct comparisons of frequentist and Bayesian approaches are difficult and often ill advised, we wished to show that BayesRR-RC provides accurate effect size estimation in the presence of LD. We provide three simple comparable metrics to assess model performance of BayesRR-RC against frequentist mixed linear association models (MLMA) applied as two-stage approaches, where either the SNP is fitted twice as it is included in both the fixed and random terms (MLMAi implemented in GCTA), or the SNP to be tested as fixed is removed from the random term alongside those on the same chromosome (MLMA implemented in BoltLMM and Regenie).

First, we calculated z-scores of the marker effect estimates from their true simulated value. As MLMA approaches estimate marker effects one-at-a-time, we calculated the z-score of the estimate from the true simulated value for the causal variants in each simulation replicate, across generative genetic models. For the Bayesian methods, at any one iteration, LD among the markers is controlled for (see Supplementary Note 4). However across iterations as the chain mixes, markers in LD will enter and leave the model, with their posterior inclusion probabilities reflecting their association with the trait. Thus, we summed the squared regression coefficient estimates of SNPs in the model at each iteration for each LD block (markers in LD $R^2 \geq 0.1$ within 1 MB) of each simulation replicate, took the posterior mean across iterations, and then calculated the z-score of the estimate from the simulated value. This metric provides an assessment of the ability of BayesRR-RC to accurately estimate the contribution of a genomic region to the phenotypic variance as compared to MLMA approaches. We present these results in Supplementary Fig. 2, where we find that the z-scores of the estimated BayesRR-RC effects are generally stable across generative genetic models and comparable to those obtained from BayesR but with slightly elevated variance in many cases as the model is less sparse (Supplementary Fig. 2a). We find that SNP effect size estimates from MLMA models have higher estimation error, especially when the causal variant is rare, or in high-LD with many other SNPs (Supplementary Fig. 2a). MLMAi models show lower estimation error than MLMA approaches, likely as they control for both distant and local LD (Supplementary Fig. 2a). We explore this further in Supplementary Note 4.

Second, to further test our PPWV approach we calculated precision-recall curves, where associations are defined as LD blocks with PPWV of ≥95% at 0.001% proportion of variance explained. True positives were the number of identified 5000 or 10,000 LD blocks that contained a causal variant. False positives were the number of identified LD blocks that did not contain a causal variant. Precision was defined as the ratio of true positives to the sum of true positives and false positives. Recall was defined as the ratio of true positives to the sum of true positives plus false negatives. The FDR was defined as the proportion of LD blocks with PPWV of ≥95% at 0.001% proportion of variance explained that did not contain a causal variant. For the MLMA methods, following standard practice, we clumped the marker effect estimates using Plink, as local LD is not controlled for, selecting LD independent markers (LD $R^2 \leq 0.01$ with other markers) across the genome. True associations were defined as selected SNPs that were in LD with a simulated causal variant (LD $R^2 \geq 0.01$). False associations were defined as selected SNPs that were not in LD (LD $R^2 \leq 0.01$) with a simulated causal variant. Precision and recall were calculated across thresholds of the chi-squared statistics of the selected markers, and the area under the curve was calculated using the trapezoid rule for calculating the integrals, assuming the curve is linear between the points. FDR is then calculated as the proportion of markers with $p$-value $\leq 5 \times 10^{-8}$ that were not in LD with a causal variant (LD $R^2 \geq 0.01$). This provides a way to directly compare model performance for the discovery of associated genomic regions across Bayesian and frequentist approaches and tests our hypothesis that a PPWV approach controls FDR well in comparison with Bonferroni $p$-value correction (Supplementary Fig. 2b, c). For both MLMA and Bayesian approaches our definition of FDR is not strictly the FDR. Markers in LD $R^2 \leq 0.01$ with the clumped selected markers may still show a weak correlation with the simulated causal variants, and likewise blocks of SNPs in LD $R^2 \leq 0.1$ may still be in weak LD with the causal variants. Our approach instead captures the ability of MLMA and Bayesian approaches to localise an effect within $R^2 \geq 0.01$ and $R^2 \geq 0.1$ respectively. We present these results in Supplementary Fig. 2.

Third, we wished to determine the out-of-sample phenotypic prediction performance of BayesRR-RC. We used the same randomly selected 10,000 individuals from the UK Biobank that were unrelated to those used in the simulation. Using the same SNP markers and the simulated marker effects we calculated a simulated genetic value for each individual across the replicates. Then, using the effects generated by BayesR and BayesRR-RC, we calculated the predicted genetic value and determined the correlation with the simulated genetic value. We took the marker effect estimates from the MLMA approaches and conducted LD clumping with p-value thresholding using Plink to find the set of markers that gave the highest correlation of the genetic predictor and the simulated genetic value

within the 10,000 UK Biobank individual selected for out-of-sample prediction. We also used the MegaPRS methods implemented in the software LDAK running the four different models described above. We compared the correlation of predicted and simulated genetic value across approaches for each of the 400 simulated phenotypes (Supplementary Fig. 2d).

*The influence of population structure and relatedness.* We then investigated the importance of controlling for multicollinearity for the control of population genetic and data structure effects. In principle, a MLMA approach will control for bias with correlated markers (either local or long-range LD) fitted as random when testing for the effects of a focal SNP. For two markers, $\mathbf{X}_1$ and $\mathbf{X}_2$ in LD correlation $\rho_{\mathbf{X}_1,\mathbf{X}_2}$, with $\beta_2 = 0$ we can express the MLMA fixed effect solution as a partial regression coefficient of the phenotype regressed onto the focal SNP after adjusting for $\mathbf{X}_2$ estimated as $u_{\mathbf{X}_2} = \frac{\mathbf{X}_2^T y}{\mathbf{X}_2^T \mathbf{X}_2 + \lambda \mathbf{I}}$. Following our derivation above for a shrinkage estimator of a partial regression coefficient the effect size of $\mathbf{X}_1$ is estimated as

$$\hat{\beta}_{y,X_1|X_2} = \frac{N}{\mathbf{X}_1^T \mathbf{X}_1} \times \rho_{y,X_1} - \frac{\rho_{\mathbf{X}_1\mathbf{X}_2} \frac{1}{N}\mathbf{X}_2^T y}{1 - \rho_{\mathbf{X}_1\mathbf{X}_2}}$$ 

and in this two-SNP example the bias is accounted for in the term $\frac{\rho_{\mathbf{X}_1\mathbf{X}_2} \frac{1}{N}\mathbf{X}_2^T y}{1 - \rho_{\mathbf{X}_1\mathbf{X}_2}}$ when the fixed effect is estimated. Multicollinearity acts to increase the $\sigma_G$ term of $\lambda$, reducing the denominator $\mathbf{X}_2^T \mathbf{X}_2 + \lambda \mathbf{I}$ in the estimation of $u_{\mathbf{X}_2}$, and increasing the variance of the estimates of common markers in high LD, those with the highest average $F_{ST}$.

We conducted a simulation study using real genomic data from chromosome 22 where 10,000 individuals were selected from two UK Biobank assessment centres (Glasgow and Croydon). First, causal variants were allocated to 5000 high-LD SNPs with effect sizes simulated from a normal distribution with variance proportional to the $F_{ST}$ among the two populations at each SNP. Second, we selected the same high-LD SNPs as the causal variants, but simulated effect sizes to have correlation 0.5 with the allele frequency differences of the SNPs among the two populations, and thus not only is the effect size proportional to the $F_{ST}$, but there is also directional differentiation (trait increasing loci tend to be those with higher allele frequency in Croydon, trait decreasing alleles have lower frequency in Croydon). For each of these two scenarios, we simulated 50 replicate phenotypes where the phenotypic variance attributable to the causal SNPs is 0.5, there is a phenotypic difference in which Croydon individuals have a phenotype that is 0.5 SD higher than Glasgow individuals (contributing variance 0.05), and residual variance was simulated from a normal with variance 0.45, to give a phenotype with mean of zero and variance of 1. The data were then analysed using a mixed-linear model association (MLMAi implemented in GCTA) and a grouped Bayesian dirac spike and slab models (BayesR implemented in GMRM). In the analysis, we either adjusted the phenotype by the first 20 PCs of the genetic data used in the simulation study, or we did not adjust the phenotype for the PCs, to examine the effects of this common methods of population stratification control. In a two-population scenario the leading eigenvector encapsulates the allele frequency differentiation between the populations and thus the expectation is that this should adjust for these differences when estimating the marker associations. The results are presented in Fig. S5a, where we find that an MLMA approach overestimates the variance attributable to the SNPs under all scenarios, both with and without adjustment for PCs. BayesR returns accurate estimates when the variance of the marker effects is proportional to $F_{ST}$ and underestimates the variance when there is a directional associations, with this underestimation being less severe with PC adjustment.

Finally, we also assess the influence of relatedness on the estimates obtained from a BayesR model using real genomic data from chromosome 21 and 22 (226,662 SNP markers) and 10,000 families randomly selected from the UK Biobank (26,034 individuals). Here, we selected 2000 LD blocks with a single causal SNP per block at random, where an LD block is defined as a group of SNP markers with absolute LD of at least 0.01. We assigned effect sizes to these 2000 selected SNPs, drawing them from a normal distribution with zero mean and variance 0.5/2000. We then multiplied effect sizes by the simulated marker values scaled to zero mean and unit variance to create the genetic values with variance 0.5. In addition to the genetic component, we added a common environment component to simulate effects coming from shared familial environment. We simulated four scenarios where each family was assigned the same common environment effect drawn from a normal distribution with variance 0 (no common environment), 0.1, 0.2, and 0.3. Finally, we added an environmental component simulated from a normal distribution with mean zero and variance 1 minus the genetic variance and minus the common environment variance. We analysed 20 replicates of each of the four scenarios with BayesRR-RC with six MAF-LD groups (terciles of MAF, each split into two groups based on median LD score within each MAF tercile). In Supplementary Fig. 5, we summarise 800 samples of the posterior distribution from 5000 iterations with a thin of five and removing the first 1000 iterations as burn-in. We find that the variance attributable to the SNPs, the regression coefficients and the posterior probability of window variance (PPWV) remain unchanged with relatedness and with increasing family effects.

*Localisation and fine-mapping of SNP-phenotype associations.* We further validate the use of PPWV in an another simulation study with 500 replicate data sets of 10,000 SNP markers for 5000 individuals for each of two scenarios. In the first scenario, 1000 SNPs are randomly selected to be causal variants and all 10,000 SNP markers are LD independent. In the second, the 1000 causal variants are each in LD

with four other variants with LD = 0.95, with the remaining 5000 variants having zero effect size and LD = 0. For each scenario, we simulate effect sizes as an equally spaced sequence from an effect size of −0.04 SD, to 0.04 SD giving genetic variance of 0.55, and we simulate residual variance from a normal distribution with zero mean and variance 0.45, to give a phenotype with zero mean and unit variance. For the first scenario, we calculate the posterior inclusion probability of each causal SNP. For the second scenario, we calculate the PPWV for each 5-SNP group. Across the 500 replicates of each scenario, we take the mean PPWV and mean PIP for each of the 1000 different effect sizes and compare these in Fig. S6a. Additionally, we grouped SNPs in 50kb regions and selected the number of regions that explain at least 0.1, 0.01 and 0.001% of the variance attributed to all SNP markers in 0.8–100% of the iterations using the simulated data described above for the multiple group enrichment scenario for chromosome 22 in the UK Biobank. We then calculated the false discovery rate (FDR), defined as the proportion of 50 kb regions identified that do not contain a causal variant, at PPWV thresholds ranging from 0.8 to 100%. We compare these in Supplementary Fig. 6b where as we lower the PPWV variance threshold, the number of false discoveries in the model increases but remains at ≤5% when the PPWV is ≥95%. This further demonstrates that our proposed PPWV approach is an appropriate metric of summarising the posterior distribution to identify associated genomic regions, irrespective of the genomic region used.

We also focused on the ability of our approach to fine-map associated regions to identify candidate SNPs and to provide a probabilistic assessment of the most likely associated set of SNP markers. To do this we used our large-scale simulation data and focused on seven focal regions within a blocks of chromosome 22. We allocated effect sizes to the following SNPs: rs131529 with MAF 0.32 which had LD $R^2 \geq 0.15$ with 348 other SNPs, rs2096537 with MAF 0.14 which had LD $R^2 \geq 0.15$ with 295 other SNPs, rs131538 with MAF 0.05 which had LD $R^2 \geq 0.15$ with 82 other SNPs, rs141962840 with MAF 0.007 which had LD $R^2 \geq 0.15$ with 11 other SNPs, rs117873986 with MAF 0.02 which had LD $R^2 \geq 0.15$ with 12 other SNPs, rs9606483 with MAF 0.005 which had LD $R^2 \geq 0.15$ with 1 other SNP, and rs78881648 with MAF 0.009 which had LD $R^2 \geq 0.15$ with 1 other SNP. To these seven SNPs, we assigned the same effect sizes in four different scenarios, either 0.05, 0.025, 0.0125, or 0.01 on the SD scale. On the remainder of chromosomes 19, 20, 21 and 22, we randomly selected 1000 SNPs as causal variants to give a polygenic background, sampling their effects from a normal distribution with zero mean and variance 0.5/1000. We repeated each of the four scenarios 20 times. We selected these regions to compare the performance of BayesRR-RC to the fine-mapping approach SuSiE as outlined in a recent paper[22]. For BayesRR-RC, we calculate the PPWV of the LD blocks containing the seven focal SNPs, and then prune these blocks based on the LD among the markers in the block (described as "purity" in the SuSiE paper[22]) to identify a credible set with LD $R^2 \geq 0.9$. We then count the proportion of times across the simulations that each causal variant was contained with one of the credible sets. For SuSiE, we ran the model from the individual-level data of the whole block of chromosome 22 using the suggested settings and setting $K = 10$. We then calculate the proportion of times that the identified credible sets contained one of the seven causal variants. We present these results in Supplementary Fig. 6c.

**UK Biobank data.** We restricted our discovery analysis of the UK Biobank to a sample of European-ancestry individuals. To infer ancestry, we used both self-reported ethnic background (UK Biobank data code 21000-0) selecting coding 1 and genetic ethnicity (UK Biobank data code 22006-0) selecting coding 1. We also took the 488,377 genotyped participants and projected them onto the first two genotypic principal components (PC) calculated from 2504 individuals of the 1000 Genomes project with known ancestries. Using the obtained PC loadings, we then assigned each participant to the closest population in the 1000 Genomes data: European, African, East-Asian, South-Asian or Admixed, selecting individuals with PC1 projection < absolute value 4 and PC 2 projection < absolute value 3. This gave a sample size of 456,426 individuals.

To facilitate contrasting the genetic basis of different phenotypes, we then removed closely related individuals as identified in the UK Biobank data release. While the BayesRR model can accommodate relatedness similar to mixed linear models, we wished to simply compare phenotypes at markers that enter the model due to LD with underlying causal variants. Relatedness leads to the addition of markers within the model to capture the phenotypic covariance of closely related individuals, and this will vary across traits in accordance with the genetic and environmental covariance for each phenotype. For these unrelated individuals, we used the imputed autosomal genotype data of the UK Biobank provided as part of the data release. We used the genotype probabilities to hard-call the genotypes for variants with an imputation quality score above 0.3. The hard-call-threshold was 0.1, setting the genotypes with probability ≤0.9 as missing. From the good quality markers (with missingness less than 5% and *p*-value for Hardy–Weinberg test larger than 10-6, as determined in the set of unrelated Europeans) were selected those with minor allele frequency (MAF) > 0.0002 and rs identifier, in the set of European-ancestry participants, providing a data set 9,144,511 SNPs, short indels and large structural variants. From these, we took the overlap with the Estonian Genome centre data to give a final set of 8,430,446 markers. From the UK Biobank European data set, samples were excluded if in the UKB quality control procedures they (i) were identified as extreme heterozygosity or missing genotype outliers; (ii)

had a genetically inferred gender that did not match the self-reported gender; (iii) were identified to have putative sex chromosome aneuploidy; (iv) were excluded from kinship inference. Information on individuals who had withdrawn their consent for their data to be used was also removed. These filters resulted in a data set with 382,466 individuals.

We then selected the recorded measures of BMI (UK Biobank variable identifier 21001-0.0) and height (variable identifier 50-0.0) collected during initial assessment visit (year 2006-2010). BMI and height phenotypes six standard deviations (SD) away from the mean were not included in the analyses. For Type 2 Diabetes (T2D) in UKB, we selected cases very broadly as individuals who have main or secondary diagnosis (UKB fields 41202-0.0–41202-0.379 and 41204-0.0–41204-0.434) of "non-insulin-dependent diabetes mellitus" (ICD 10 code E11) or self-reported non-cancer illness (UKB field 20002-0.0–20002-2.28) "type 2 diabetes" (code 1223). From respondents self-reporting just "diabetes" (code 1220), we selected as cases those who did not self-report "type 1 diabetes" (code 1222) and had no Type 1 Diabetes (ICD code E10) diagnosis. Individuals with self-reported "diabetes" and "type 1 diabetes"/E10 were also left out from controls. We also defined coronary artery disease (CAD) cases broadly as participants with one of the following primary or secondary diagnoses or cause of death: ICD 10 codes I20 to I28; self-reported angina (code 1074) or self-reported heart attack/myocardial infarction (code 1075). Participants with self-reported "heart/cardiac problem" (code 1066) were not included as cases but also excluded from controls. This gave a sample size for each trait of 25,773 T2D cases and 359,730 T2D controls, 39,766 CAD cases and 344,054 CAD controls, 382,402 measures of height and 381,899 measures of BMI.

UK Biobank has approval from the North West Multi-centre Research Ethics Committee (MREC) to obtain and disseminate data and samples from the participants (http://www.ukbiobank.ac.uk/ethics/), and these ethical regulations cover the work in this study. Written informed consent was obtained from all participants. Data from this project were held under UK Biobank project ID 35520.

All phenotypes were adjusted for age of attending assessment centre (UKB code 21003-0.0, factor with levels for each age), year of birth (UKB field 34-0.0, factor with levels for each year), UK Biobank recruitment centre (UKB field 54-0.0, factor with levels for each centre), Genotype batch (UKB field 22000, factor with levels for each batch) and final 20 leading principal components of 1.2 million LD clumped markers from the 8,430,446 markers included in the analysis, calculated using flashPCA (see "Code availability" section). The residuals were then converted to z-scores with 0 mean and variance of 1. Similarly as for relatedness, population stratification is also accounted for within the BayesRR model through the addition of a background of marker effects entering the model, however we also wished to account for this in the standard manner by adjusting for the leading 20 PCs of the SNP data to get as close as possible to the inclusion of markers in the model that reflect LD with the causal variants. We note that as with any association model, while we take steps to adjust for known spatial (UKB centre), batch, and ancestry effects, and that the effects of each SNP is estimated jointly (and thus conditionally on the effects of all the other SNPs) environmentally induced covariance between SNP markers and a phenotype is still possible.

We partition SNP markers into seven location annotations using the knownGene table from the UCSC browser data (see "Code availability" section). We preferentially assigned SNPs to coding (exonic) regions first, then in the remaining SNPs, we preferentially assigned them to intronic regions, then to 1 kb upstream regions, then to 1–10 kb regions, then to 10–500 kb regions, then to 500–1 Mb regions. Remaining SNPs were grouped in a category labelled "others" and also included in the model so that variance is partitioned relative to these also. Thus, we assigned SNPs to their closest upstream region, for example if a SNP is 1 kb upstream of gene X, but also 10–500 kb upstream of gene Y and 5 kb downstream for gene Z, then it was assigned to be a 1 kb region SNP. This means that SNPs 10–500 kb and 500 kb–1 Mb upstream are distal from any known nearby genes. We further partition upstream regions to experimentally validated promoters, transcription factor binding sites (tfbs) and enhancers (enh) using the HACER, snp2tfbs databases (see "Code availability" section). All SNP markers assigned to 1 kb regions map to promoters; 1–10 kb SNPs, 10–500 kb SNPs, 500 kb–1 Mb SNPs are split into enh, tfbs and others (un-mapped SNPs) extending the model to 13 annotation groups. Within each of these annotations, we have three minor allele frequency groups (MAF < 0.01, 0.01 > MAF > 0.05, and MAF > 0.05), and then each MAF group is further split into two based on median LD score. This gives 78 non-overlapping groups for which our BayesRR-RC model jointly estimates the phenotypic variation attributable to, and the SNP marker effects within, each group. For each of the 78 groups, SNPs are modelled using five mixture groups with variance equal to the phenotypic variance attributable to the group multiplied by constants (mixture 0 = 0, mixture 1 = 0.0001, 2 = 0.001, 3 = 0.01, 4 = 0.1). We conducted a series of convergence diagnostic analyses of the posterior distributions to ensure we obtained estimates from a converged set of four Gibbs chains, each run for 6000 iterations with a thin of five and burn-in of 500 for each trait (Supplementary Figs. 7–10).

We calculate PPWV for LD blocks of the genome, by first calculating the minor allele frequency of each SNP (p) and using 1 − p in a Plink clumping procedure to select LD independent (correlation² ≤ 0.1) blocks of SNPs. We then repeat the estimation of the PPWV of 50 kb regions across the genome, then map SNPs to the coding region of genes, and to the closest gene +/− 50 kb from the SNP position. These are labelled as located in a coding region, an intron, 1 kb upstream of a gene using our functional annotations. Remaining SNPs are labelled as located in a cis-

region (up to +/−50 kb from a gene, Supplementary Data 6–9). Finally, we mapped SNPs with greater than 50% posterior inclusion probability (PIP) across all four chains labelling them using our seven location annotations (Supplementary Fig. 13). We report SNPs with PIP > 95% and their corresponding p-value from reported GWAS summary statistics (fastGWA, see "Code availability") with "body mass index" entry for HT, "standing height" for HT, "angina/heart attack" for CAD and "diabetes" for T2D (Supplementary Data 10).

We then compared our BayesRR-RC estimates for height, BMI, T2D and CAD to RHEmc[18] which also relies on individual level data. We ran RHEmc with ten independent random vectors and 100 jackknife blocks on the 382,466 individuals and 8,430,446 SNP markers assigned to our 78 non-overlapping groups. SNP heritability estimates, enrichment and standard errors per genetic component are reported in Supplementary Data 3. We intended to include SNP heritability estimates from Bolt-REML[17] in the method comparison but the run time and memory usage exceeded 7 days and 900 GB which is the limiting run-time and memory for our HPC system. Among the summary statistic methods, we ran sLDSC[19] and SumHer[6]. To do so, we created summary statistics containing marginal associations for each of the 8,430,446 markers using plink2[28] for height, BMI, T2D and CAD. For sLDSC, we computed univariate LD scores and annotation-specific LD scores for the 78 non-overlapping groups using a window size of 10,000 kb and a subset of 20,000 individuals randomly selected from the full data set. We partitioned heritability with our annotations and no restriction on MAF. SNP heritability estimates, proportions of heritability, enrichment and standard errors per genetic component are reported in Supplementary Data 4. For SumHer, we computed LDAK weightings and created tagging files separately by chromosomes using the full data set (M = 8,430,446 and N = 382,466) as reference and a window size of 1000 kb. Because SNPs included in groups others and rare 1Mb tfbs are not present in all chromosomes, tagging files are constructed using 70 non-overlapping annotations only. The remaining SNPs are modelled together in an extra partition. Finally, we merged the tagging files and regressed the summary statistics onto this file assuming the LDAK model. SNP heritability estimates, proportions of heritability, enrichment and standard errors per genetic component are reported in Supplementary Data Table 5. The proportion of genetic variance estimated genome-wide with RHE-mc, sLDSC, and SumHer are shown in Table 1. We also report the proportion of genetic variance attributed to SNPs located in exons, introns, 1, 1–10 and 10–500 kb regions and the proportion of common SNPs located in exons, introns and 10–500 kb regions computed from the single heritability estimates observed (Table 1).

In addition to plink2[28] summary statistics, we also applied Bolt-LMM[8] and Regenie[9] to height, BMI, T2D and CAD. In the first step, we pruned SNPs using plink[30] with a pairwise r2 threshold of 0.5 and a window size of 1000 kb, resulting in a subset of 1,362,013 SNPs markers. We restricted the random effects in the mixed model for bolt-LMM and the ridge regression predictors for Regenie to this subset of pruned SNPs. In the second step, all 8,430,446 SNPs from the full genotype data were then tested for association in both methods. Following recommendations, we used the provided hg19 genetic map file and 1000 Genomes LD scores reference for Bolt-LMM and performed the default mixed linear model association test. For Regenie, the 1,362,013 SNP markers are split in blocks of 1000 consecutive SNP markers and ridge regression predictors are computed for a range of five shrinkage parameters within each block. For the association testing, we split the 8,430,446 SNP markers in blocks of 400 consecutive SNP markers and set the Firth correction p-value threshold to 0.01. We then applied an approximate and joint association analysis called GCTA-COJO[31] to the summary statistics obtained with Bolt-LMM, Regenie and plink2. We ran GCTA-COJO using a subset of 20,000 individuals randomly selected from the 382,466 individuals as reference with a window size of 10,000 kb and a r2 cutoff value of 0.5 for the LD among the SNPs in the data. Finally, we set a p-value threshold to 5e−8 to report significant SNPs associated with height, BMI, CAD an T2D in Table 2.

**Estonian Genome Centre data.** For the Estonian Genome Centre Data, 32,594 individuals were genotyped on Illumina Global Screening (GSA) arrays and we imputed the data set to an Estonian reference, created from the whole genome sequence data of 2244 participants[32]. From 11,130,313 markers with imputation quality score >0.3, we selected SNPs that overlapped with the UK Biobank, resulting in a set of 8,433,421 markers.

We selected height and BMI measures from the Estonian Genome Centre data, in 32,594 individuals genotyped on GSA array and converted them to sex-specific z-scores after applying the same outlier removal procedure as in UKB and adjusting for the age at agreement. Prevalent cases of CAD and T2D in the Estonian Biobank cohort were first identified on the basis of the baseline data collected at recruitment, where the information on prevalent diseases was either retrieved from medical records or self-reported by the participant. The cohort was subsequently linked to the Estonian Health Insurance database that provided additional information on prevalent cases (diagnoses confirmed before the date of recruitment) as well as on incident cases during the follow-up.

All Estonian biobank participants have signed a broad informed consent form and the study was carried out under ethical approval 1.1 12/2856 from the Estonian Committee on Bioethics and Human Research (Estonian Ministry of Social Affairs).

As the UK Biobank marker effects are estimated from traits that were standardised to a z-score prior to analysis, all effect sizes obtained are on the SD

scale. Thus when we create a genomic predictor, for say coding SNPs, by multiplying SNPs mapped to coding regions genotyped in Estonia to the effect sizes obtained in the UK Biobank for each iteration, we obtain a genetic predictor for each iteration, providing a posterior predictive distribution that is also on the SD scale. For each trait, we created 2000 genomic predictors for each individual in the Estonian Biobank, at each of the 13 annotation groups, by selecting effect size estimates obtained every tenth iteration from the last 3000 iterations of each of the four Gibbs chains and combining them together in a single posterior. We calculated prediction accuracy as the proportion of phenotypic variation explained by the genomic predictor, and area under the receiver operator curve (AUC) for T2D and CAD using each individual's mean genetic predictor. For each of the 13 annotation groups, we calculated the partial correlation of the genetic predictor of each of the 2000 iterations and the phenotype. We then used this to estimate the independent proportional contribution of each group to the total prediction accuracy, providing a metric of replication for our UK Biobank enrichment results.

For height and BMI, we determined the probability that each Estonian individual's predictor accurately reflected their phenotypic value. To do this, we calculated the proportion of posterior samples with abs $(\hat{g} - y)$ of less than 1 for each individual, which gives a measure of the degree to which each posterior predictive distribution overlaps with the phenotype within $+/-1$ SD.

For T2D and CAD, we extended the PCF metric, typically defined as the proportion of cases with larger estimated risk than the top $p^{th}$ percentile of the distribution of genetic risk in the general population. We calculated the proportion of posterior samples for each individual with values in the top 25% of the distribution of genomic predictors for each trait. Thus for each individual, we calculate the probability that the posterior predictive distribution is in the top 25% of the distribution of genetic risk in the general population.

As a comparison, we also estimated a boltLMM prediction model using MegaPRS[21] as recommended by the authors and as shown to have the best prediction performance out of the MegaPRS approaches in our simulation study. We clumped SNPs with r2 threshold of 0.5 resulting in 1,508,624 SNP markers to be included in the analysis and randomly selected 20,000 individuals to compute the LDAK weights. We then computed the tagging file using the same data set as reference and the 64 BLD-LDAK annotations. Here, weights are models as an extra annotation and we save the heritability matrix. We then regress the plink2[28] summary statistics for height, BMI, CAD and T2D onto the tagging file, saving the per-predictor heritabilities. We then created four reference panels with the same 1,508,624 SNP markers but randomly selecting different 5000 related individuals from the UK Biobank and we used these to: (i) calculate predictor-predictor correlations with a window size of 3000 kb to estimate the LD structure; (ii) compute pseudo summaries from the plink2 summary statistics including ambiguous alleles, which creates pseudo training and test summary statistics to be used in the construction of the prediction model; (iii) estimate effect sizes specifying a Bolt-LMM model for height, BMI, CAD and T2D, using the predictor–predictor correlations, the per-predictor heritabilities, the plink2 summary statistics and training pseudo summary statistics, whilst including ambiguous allele and specifying a 1000 kb window; (iv) test prior distributions to determine the most accurate model and obtain the best effect sizes. These steps resulted in 1,397,514 predictors for height, 1,471,586 for BMI, 1,397,514 for CAD and 1,389,364 for T2D and we ensured that at no point was the Estonian genome centre data used, nor was any overlapping individuals in the UK Biobank subsets used to train the models and the data used to generate the summary statistics. Finally, we then calculated genomic predictors for each individual in the Estonian Biobank using the best effect sizes. We report the squared correlations between the genomic predictor and phenotypes.

**Reporting summary**. Further information on research design is available in the Nature Research Reporting Summary linked to this article.

## Data availability
This project uses UK Biobank data under project 35520. The Estonian Genome Centre data are protected and are not available due to data privacy laws. The Estonian Genome Centre data can be made available under restricted access upon request from the cohort author R.M. with appropriate research agreements. Summaries of all posterior distributions generated in this study are provided in Supplementary Data tables. Full posterior distributions of the SNP marker effects sizes and estimated variance components for each trait are deposited on Dryad with https://doi.org/10.5061/dryad.sqv9s4n51.

## Code availability
Our BayesRR-RC model is implemented within the software GMRM, with full open source code available at: https://github.com/medical-genomics-group/gmrm. UCSC Table Browser https://genome.ucsc.edu/cgi-bin/hgTables. flashPCA https://github.com/gabraham/flashpca. Plink1.90 https://www.cog-genomics.org/plink2/. GCTA https://cnsgenomics.com/content/software. HACER database http://bioinfo.vanderbilt.edu/AE/HACER/. snp2tfbs database https://ccg.epfl.ch/snp2tfbs/. fastGWA database http://fastgwa.info/ukbimp/phenotypes/. Computing environment https://www.epfl.ch/research/facilities/scitas/hardware/helvetios/.

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

## Acknowledgements

This project was funded by an SNSF Eccellenza Grant to MRR (PCEGP3-181181), and by core funding from the Institute of Science and Technology Austria. We would like to thank the participants of the cohort studies, and the Ecole Polytechnique Federal Lausanne (EPFL) SCITAS for their excellent compute resources, their generosity with their time and the kindness of their support. P.M.V. acknowledges funding from the Australian National Health and Medical Research Council (1113400) and the Australian Research Council (FL180100072). L.R. acknowledges funding from the Kjell & Märta Beijer Foundation (Stockholm, Sweden). We also would like to acknowledge Simone Rubinacci, Oliver Delanau, Alexander Terenin, Eleonora Porcu, and Mike Goddard for their useful comments and suggestions.

## Author contributions

M.R.R. conceived and designed the study. M.P., D.T.B. and A.K. contributed to the study design. M.P. and M.R.R. conducted the experiments and analyses with input from D.T.B., A.K., S.E.O., A.H., J.S., P.M.V., R.M. and L.R. M.R.R., D.T.B., S.E.O. and L.R. derived the equations and the algorithm. EJO and DTB developed the software, with contributions from M.R.R., M.P., S.E.O., A.K. and G.M. M.R.R., M.P. and DTB wrote the paper. RM and ZK provided study oversight and contributed data to the analysis. All authors approved the final manuscript prior to submission.

## Competing interests

The authors declare no competing interests.
