## [Peer Review File · Nature Communications]

Probabilistic inference of the genetic architecture of functional enrichment of complex traitsEditorial Note: This manuscript has been previously reviewed at another journal that is not operating a transparent peer review scheme. This document only contains reviewer comments and rebuttal letters for versions considered at Nature Communications.

Reviewers' Comments:

Reviewer #2:

Remarks to the Author:

The authors responded to my concern regarding the standard of association summary statistics. However, I think the debate is not resolved yet.

The authors keep emphasizing that the single model is able to conduct loci discovery, control the FDR, and also provide more power. This is not addressing the concern. The problem here is that a substantial amount of GWAS has already been conducted and published with genome-wide summary association statistics. Many methods have been developed to take advantage of the new type of data, i.e. summary statistics. In many cases, we do not have access to individual-level data, while the available summary statistics can bring a lot of power to new studies. In terms of power, summary-stats-based methods can still beat the new model as long as the sample size that generated the summary statistics is much larger, which is often the case. There is also the data security and availability issue for individual-level data.

Furthermore, for the discussion on statistics frameworks. This has nothing to do with the history of Bayesian methods. All the information regarding the statistical inference is available in the likelihood, and the p-value is simply a useful statistic for each single hypothesis test. There is always the sensitivity-specificity trade-off in any multiplicity scientific discovery problem. What's useful in the standard GWAS summary statistics is the nature of the estimated marginal genetic effects, with proper standard errors, on which downstream analysis models could be developed, making use of various summary statistics resources in an efficient and flexible way.

With these, I believe it is wrong to state that the new model can completely replace standard GWAS. I suggest the authors be very clear about the pros and cons of their "discovery metrics" and standard GWAS summary statistics, perhaps mention a few state-of-the-art methods based on GWAS summary statistics, and discuss this issue carefully.

Reviewer #3:

Remarks to the Author:

In this manuscript, the authors present a Bayesian method to perform statistical inference on individual-level GWAS data. The most important novelty is that it combines many inference tasks that are often separated into a single estimation task. This approach is appealing. In addition, the model that is fit by the method is quite flexible and complicated, which may have advantages when compared with methods that use simpler models.

Since it appears this manuscript has already been through multiple rounds of review, my review will be restricted to the concerns that have been raised by other reviewers, in particular Reviewer #1. I'll discuss three of the Reviewer's main concerns.

First, the reviewer was concerned about the motivation for a single-step approach to multiple

estimation tasks. The reviewer points out that the predominant two-step approach, which involves an initial regression or mixed-model association step followed by statistical inference (h² components, fine mapping and risk prediction) on the summary statistics, has major advantages. In my opinion, this criticism should not preclude publication, as it is important for high-impact journals to publish alternative approaches that have advantages over commonly-used pipelines, even if they also have disadvantages. However, I do think the authors need to incorporate the Reviewer's points into their manuscript, and overall, the paper would be much more compelling if there were more discussion about how these analyses are usually done, why it could be advantageous to integrate them into a single step, and why some users may prefer to use a multi-step approach instead. There are the beginnings of this discussion in lines 36-44, but currently this paragraph is weak. It gave me the wrong impression that BayesRR-RC was just a functionally-informed mixed model association method.

Second, the reviewer was concerned about the posterior causal effect size estimates, which differ from marginal association statistics. Reviewer #2 also requested clarification about this point. Reviewer #1 correctly points out that (a) marginal statistics are useful for downstream analyses, and (b) they are easier to interpret, as they don't depend on the particular model that is being used. The authors respond to (a) with the dubious claim that their effect-size estimates can be used as the input to downstream methods that input summary association statistics. I disagree with this, as the sampling distribution of the association statistics is totally different from that of the Gibbs sampler estimates. It is not a question of one or the other being better – they are different categories of things. Moreover, the authors state that their samples from the posterior could be used as input to a downstream method in order to avoid using a jackknife to compute standard errors. Again, I don't agree with this, and I don't recommend including it in the manuscript.

They respond to (b) by arguing that marginal association statistics, if computed using mixed models, are also affected by the model that is used. This is technically true, but MLM marginal test statistics are much less sensitive to the prior than posterior effect sizes, as the prior is being applied to the effect sizes of *other* SNPs with the focal SNP being treated as a fixed effect. I recommend that the authors think carefully about how they are presenting their effect size estimates and try to avoid confusing readers about how they compare with marginal association statistics.

Third, the reviewer was concerned that the simulations are unrealistically favorable to the method under consideration. In particular, they are worried that these simulations are well powered, and that the method with its very flexible prior will perform well specifically when there is enough power to estimate the large number of parameters that go into the model. I share this concern, and I'm not convinced by the authors' assertion that the simulations are actually *not* well powered. For one thing, in Figure 1a, it appears that bayesRR-RC obtains highly accurate estimates in every simulation that is considered, so it's hard to argue that these simulations are extremely challenging. For another, the authors don't vary the sample size in these simulations, nor do they vary the heritability, and they barely vary the polygenicity (5000 vs 10000 causal SNPs being a tiny difference compared with the range that is observed across real traits; I usually consider a range of 100 or more for the polygenicity parameter in simulations).

It seems the authors have spent a lot of time and energy trying different MAF/LD-dependent architectures, but it would probably suffice to focus on a much smaller number of *realistic* MAF/LD architectures, and move unrealistic architectures, for example those with higher causal effect sizes for high-LD SNPs, to the supplement. (Also, is "b" in Table 1 a per-allele effect size or a per-normalized-genotype effect size? It would suffice to consider architectures where the per-allele effect size is larger for rare SNPs, but the per-normalized-genotype effect size is larger for common SNPs). It's hard to ask for more simulations when the authors have already done so many, but this is really a question of quality, not quantity – currently all the simulations are devoted to exploring every possible

permutation of these MAF/LD parameters, while neglecting parameters (polygenicity and sample size/heritability) that vary widely across traits/datasets in practice. I suggest focusing the main simulations on 1-2 realistic values for MAF/LD-dependent architecture and trying 2-3 different values of polygenicity that differ by a factor of 10-100, as well as 2-3 different values of sample size/h² that differ by a similar margin. It is not necessary that these simulations show that BayesRR-RC is unambiguously the most accurate method out of all of those tested, and it would be good if the most underpowered simulations show BayesRR-RC producing very noisy estimates, so that we can have some understanding of the sample size/h² that is needed for it to perform well.

Finally, I strongly agree with the Reviewer that Figures 1-2 contain too much information. I count 140 subpanels in Figure 1, and as if that weren't enough, 200 in Figure 2! Frankly, when I encounter a figure like this in a paper, I stop reading. It should be possible to a large number of analyses but then distill them down to a readable figure.

Signed,
Luke O'Connor

We thank both Reviewers for their comments, which we feel have greatly improved our manuscript. We have followed all of the suggestions made, and substantially revised our work in the hope of addressing all concerns raised.

As an overview, we have:

- 1. Conducted a further large-scale simulation analysis of 180 simulated phenotypes, creating 18 different polygenicity and sample size settings to ensure that we now cover a wide-range of power scenarios, as well as different MAF-LD architectures. The results are presented in Figure 1 and 2, moving the previous results based on 400 simulated phenotypes to Supplementary Figures S1 to S4.**
- 2. We have revised the discussion to provide a more comprehensive discussion of the limitations given previous GWAS data availability (lines 360-393).**
- 3. We have modified the introduction to clarify and better describe what is being estimated and how our model differs to commonly used approaches (lines 20-54).**
- 4. We have restructured the Results section to describe the new simulation work and to make it also more concise and clear.**
- 5. We have restructured the manuscript, creating a series of four Supplementary Notes as we feel this improves the presentation and the clarity of the manuscript.**

Below, we address each of the concerns raised in full.

REVIEWER COMMENTS

Reviewer #2 (Remarks to the Author):

The authors responded to my concern regarding the standard of association summary statistics. However, I think the debate is not resolved yet.

The authors keep emphasizing that the single model is able to conduct loci discovery, control the FDR, and also provide more power. This is not addressing the concern. The problem here is that a substantial amount of GWAS has already been conducted and published with genome-wide summary association statistics. Many methods have been developed to take advantage of the new type of data, i.e. summary statistics. In many cases, we do not have access to individual-level data, while the available summary statistics can bring a lot of power to new studies. In terms of power, summary-stats-based methods can still beat the new model as long as the sample size that generated the summary statistics is much larger, which is often the case. There is also the data security and availability issue for individual-level data.

Furthermore, for the discussion on statistics frameworks. This has nothing to do with the history of Bayesian methods. All the information regarding the statistical inference is available in the likelihood, and the p-value is simply a useful statistic for each single hypothesis test. There is always the sensitivity-specificity trade-off in any multiplicity scientific discovery problem. What's useful in the standard GWAS summary statistics is the nature of the estimated marginal genetic effects, with proper standard errors, on which downstream analysis models could be developed, making use of various summary statistics resources in an efficient and flexible way.

With these, I believe it is wrong to state that the new model can completely replace standard GWAS. I suggest the authors be very clear about the pros and cons of their "discovery metrics" and standard GWAS summary statistics, perhaps mention a few state-of-the-art methods based on GWAS summary statistics, and discuss this issue carefully.

We thank the reviewer for their comments and their efforts to improve the presentation of our work. We had no intention of stating that our model can completely replace standard GWAS, and we regret that this was the impression given. Rather, we see our approach as an alternative and we now clearly state this throughout the abstract, introduction, results, and discussion. Our aim with this work is to provide a novel way to analyse the large-scale biobank studies that are becoming increasingly available and we are working on providing a way of combining posterior distributions obtained across biobank studies.

We now discuss the limitations of not providing marginal SNP associations and not being able to utilise these, as suggested by the reviewer in our Discussion on lines 360-393. We have also re-written our introduction to better contrast and compare estimation approaches (lines 20-54). Additionally, the improved simulations suggested by Reviewer #3 below, make our ability to localise SNP-phenotype association clear compared to mixed-linear model association methods.

Reviewer #3 (Remarks to the Author):

In this manuscript, the authors present a Bayesian method to perform statistical inference on individual-level GWAS data. The most important novelty is that it combines many inference tasks that are often separated into a single estimation task. This approach is appealing. In addition, the model that is fit by the method is quite flexible and complicated, which may have advantages when compared with methods that use simpler models.

Since it appears this manuscript has already been through multiple rounds of review, my review will be restricted to the concerns that have been raised by other reviewers, in particular Reviewer #1. I'll discuss three of the Reviewer's main concerns.

First, the reviewer was concerned about the motivation for a single-step approach to multiple estimation tasks. The reviewer points out that the predominant two-step approach, which involves an initial regression or mixed-model association step followed by statistical inference (h^2 components, fine mapping and risk prediction) on the summary statistics, has major advantages. In my opinion, this criticism should not preclude publication, as it is important for high-impact journals to publish alternative approaches that have advantages over commonly-used pipelines, even if they also have disadvantages. However, I do think the authors need to incorporate the Reviewer's points into their manuscript, and overall, the paper would be much more compelling if there were more discussion about how these analyses are usually done, why it could be advantageous to integrate them into a single step, and why some users may prefer to use a multi-step approach instead. There are the beginnings of this discussion in lines 36-44, but currently this paragraph is weak. It gave me the wrong impression that BayesRR-RC was just a functionally-informed mixed model association method.

We agree and we thank Reviewer #3 for their comment. We have now revised our introductory paragraphs to clarify this and to avoid giving the wrong impression. We have modified lines 20-54 and also the discussion lines 360-393 to include all of the points raised by the reviewer.

Second, the reviewer was concerned about the posterior causal effect size estimates, which differ from marginal association statistics. Reviewer #2 also requested clarification about this point. Reviewer #1 correctly points out that (a) marginal statistics are useful for downstream analyses, and (b) they are easier to interpret, as they don't depend on the particular model that is being used. The authors respond to (a) with the dubious claim that their effect-size estimates can be used as the input to downstream methods that input summary association statistics. I disagree with this, as the sampling distribution of the association statistics is totally different from that of the Gibbs sampler estimates. It is not a question of one or the other being better – they are different categories of things. Moreover, the authors state that their samples from the posterior could be used as input to a downstream method in order to avoid using a jackknife to compute standard errors. Again, I don't agree with this, and I don't recommend including it in the manuscript.

*They respond to (b) by arguing that marginal association statistics, if computed using mixed models, are also affected by the model that is used. This is technically true, but MLM marginal test statistics are much less sensitive to the prior than posterior effect sizes, as the prior is being applied to the effect sizes of *other* SNPs with the focal SNP being treated as a fixed effect. I recommend that the authors think carefully about how they are presenting their effect size estimates and try to avoid confusing readers about how they compare with marginal association statistics.*

Many thanks for this comment, we agree that our statements were not clear and we have removed this section of the discussion. We did not intend to state that our estimates can be used as they are for current downstream analysis approaches. Rather, that alternative approaches could be devised to utilise the estimates we provide, as we agree that they are different categories of things. We now leave this to future work, and we now clarify the presentation of our effect size estimates within the introduction on lines 34-50 and they compare to the standard of estimating SNP associations as fixed effects.

*Third, the reviewer was concerned that the simulations are unrealistically favorable to the method under consideration. In particular, they are worried that these simulations are well powered, and that the method with its very flexible prior will perform well specifically when there is enough power to estimate the large number of parameters that go into the model. I share this concern, and I'm not convinced by the authors' assertion that the simulations are actually *not* well powered. For one thing, in Figure 1a, it appears that bayesRR-RC obtains highly accurate estimates in every simulation that is considered, so it's hard to argue that these simulations are extremely challenging. For another, the authors don't vary the sample size in these simulations, nor do they vary the heritability, and they barely vary the polygenicity (5000 vs 10000 causal SNPs being a tiny difference compared with the range that is observed across real traits; I usually consider a range of 100 or more for the polygenicity parameter in simulations).*

*It seems the authors have spent a lot of time and energy trying different MAF/LD-dependent architectures, but it would probably suffice to focus on a much smaller number of *realistic* MAF/LD architectures, and move unrealistic architectures, for example those with higher causal effect sizes for high-LD SNPs, to the supplement. (Also, is "b" in Table 1 a per-allele effect size or a per-normalized-genotype effect size? It would suffice to*

consider architectures where the per-allele effect size is larger for rare SNPs, but the per-normalized-genotype effect size is larger for common SNPs). It's hard to ask for more simulations when the authors have already done so many, but this is really a question of quality, not quantity – currently all the simulations are devoted to exploring every possible permutation of these MAF/LD parameters, while neglecting parameters (polygenicity and sample size/heritability) that vary widely across traits/datasets in practice. I suggest focusing the main simulations on 1-2 realistic values for MAF/LD-dependent architecture and trying 2-3 different values of polygenicity that differ by a factor of 10-100, as well as 2-3 different values of sample size/h² that differ by a similar margin. It is not necessary that these simulations show that BayesRR-RC is unambiguously the most accurate method out of all of those tested, and it would be good if the most underpowered simulations show BayesRR-RC producing very noisy estimates, so that we can have some understanding of the sample size/h² that is needed for it to perform well.

We thank the reviewer for their suggestion. We agree and we have repeated our entire simulation scheme creating 18 different polygenicity and sample size settings. As suggested we selected a realistic MAF/LD architecture and kept the sample size the same at 40,000 individuals. We simulated three different levels of polygenicity, selecting either 1,000, 10,000, or 100,000 causal variants. Within each level of polygenicity, we then selected three different heritability values of 0.1, 0.3, and 0.6 which led to 9 settings in which we then assign effect sizes to markers in two different ways. We first assign them at random across genomic annotations groups. Here, the marker effects are sampled from normal distributions with variances 6e-04, 3e-04, 1e-04, 6e-05, 3e-05, 1e-05, 6e-06, 3e-06, 1e-06. When we then repeat the simulation for the enriched setting, where we vary the proportion of SNP-heritability across the 13 annotation groups, the marker effect variances differ even further.

Given a sample size of 40,000 the SE of OLS regression is $1/\sqrt{40,000} = 0.005$ so 1.96 times this would be an absolute effect size that is significantly detectable at $p=0.05$ of 0.0098, which squared is a contribution to variance of 9.604e-05. So in 6 of our simulation settings, the average effect size is less than the effect size there is power to detect and in 3 of the 18 settings both BoltLMM and plink yielded no genome-wide significant findings.

We now present the results of this simulation in Figures 1 and 2. Figure 1 shows that in low polygenicity settings BayesRR-RC performs poorly as compared to REML when estimating the variance attributable to different genomic annotation groups, with the correlation of the simulated and estimated values dropping to around 0.5, with up to 20% of the annotation groups being mislabelled as having enrichment values greater than or equal to 2. With so few causal variants the posterior distribution becomes more multimodal as effect sizes can be assigned to single markers or multiple markers within an annotation group and this appears to lead to poorer estimates of the hyperparameters. Generally, as the polygenicity increases BayesRR-RC performs similar to the REML model and both perform better than RHEmc, LDSC, and Sumher. This is in agreement with the previous simulation findings, but clearly shows situations where a REML model is preferred over BayesRR-RC. As estimating a 78 component REML model in UK Biobank data is computationally intensive with huge RAM requirements, our method offers clear advantages for large-scale biobank studies, where individual-level data are available.

In Figure 2, we first compare our PPWV approach to mixed-linear model association (MLMA) estimates obtained by Bolt-LMM. We thought the simplest comparison to present here was an estimate of power, defined as the probability within each simulation replicate of localising an effect size to a group of SNPs in LD within a 1MB region. We select only

Bolt-LMM here as the comparison to improve clarity of the figure, and as all of our other simulation work showed no meaningful differences in estimation across MLMA models. We find that Bolt outperforms BayesRR-RC for localisation of large-effect loci, but that our proposed PPWV approach localises causal variants well under high polygenicity.

Second, we then compare BayesRR-RC prediction accuracy obtained in an independent sample, to simple MLMA fixed effect estimates and four different summary statistic approaches implemented in the LDAK software. Generally, for the same sample size, BayesRR-RC yields higher prediction accuracy than the best selected LDAK model.

This simulation set-up is now described in the Methods and we have taken the advice of Reviewer #3 a step further and streamlined the presentation. Our previous Figures 1-2 are now in the Supplementary Material (Figures S1-S4) alongside our simulation work inferring the influence of population stratification and relatedness on the estimates (Figure S5), and our work comparing our Bayesian regional association mapping (the PPWV approach) to a state-of-the-art fine-mapping approach (SuSie, Figure S6). We then move the smaller-scale simulations into four Supplementary Notes, so that each small-scale simulation supports the specific part of the theory or algorithm development. Thus, we now have:

Supplementary Note 1 which outlines the basic derivation we used to arrive at our new model formulation.

Supplementary Note 2 describes the derivation of the novel hybrid-parallel BSP Gibbs sampling algorithm with the unique highly vectorised look-up table dot product estimation and here we include the simulation results testing the stability of the algorithm with increasing parallelism. These steps allow a novel form of a Gibbs sampler to scale to large-scale biobank data, and so this paper is much more than combining many inference tasks that are often separated into a single estimation task.

Supplementary Note 3 with the discussion of the PPWV approach. We agree that generally many people will prefer p-value testing, but this section provides an alternative Bayesian effects size localisation approach, that rivals state-of-the-art fine-mapping approaches.

Supplementary Note 4 with the derivation of the model under multicollinearity and the simulation results comparing the sparsity inducing prior with other statistical models. This work helps explain why a model with MAF-LD separated hyperparameters with sparsity inducing priors performs well in correlated SNP data.

Finally, I strongly agree with the Reviewer that Figures 1-2 contain too much information. I count 140 subpanels in Figure 1, and as if that weren't enough, 200 in Figure 2! Frankly, when I encounter a figure like this in a paper, I stop reading. It should be possible to a large number of analyses but then distill them down to a readable figure.

We have revised Figure 1 and Figure 2 to present the results from the simulation study suggested by the reviewer.

We move our previous Figures 1-2 to the Supplementary Material and we expand them into four separate Supplementary Figures S1-S4, whilst simplifying their presentation. We hope this is fine for the Supplementary Figures as we want to be able to compare the

different MAF/LD-dependent architectures across approaches and present the full information.

Reviewers' Comments:

Reviewer #2:

Remarks to the Author:

The authors have addressed my major concerns properly. Also, the new Figure 1 and Figure 2 are much better presentations than the old versions.

Reviewer #3:

Remarks to the Author:

The authors have addressed my comments.